# A selective enrichment and specific probe terminal mediated strategy for highly sensitive detection of microRNAs

Zecheng Zhong[1,2,11], Weida Huang[1,2,11], Jinhua Ren[3,4,5,6,11], Li Yuan[7,8,11], Qiurong Zhong[9,11], Yi Ge[1,2,11], Jin Wang[1,2], Jiyu Xiang[1,2], Lesi Lin[1,2], Yanmei Chen[9], Jingjing Xu[3,4,5], Ziyao Zhang[7,8], Tingdong Li[1,2], Jun Zhang[1,2], Jianda Hu[5,10], Shuizhen He[9] ✉, Linquan Tang[7,8] ✉, Shengxiang Ge[1,2] ✉, Ningshao Xia[1,2] ✉, Ting Yang[3,4,5,6] ✉ & Shiyin Zhang[1,2] ✉

MicroRNAs are promising liquid biopsy biomarkers, but their clinical translation is hindered by detection challenges, including low abundance, high sequence similarity, and background interference. Here we present SE-SPTM-PCR, a detection platform integrating selective miRNA enrichment using locked nucleic acid probes with specific probe terminal mediated RT-qPCR. We show that SE-SPTM-PCR eliminates nonspecific amplification and achieves 100-fold higher sensitivity than conventional stem-loop RT-qPCR. In clinical studies, SE-SPTM-PCR significantly improves hsa-miR-92a-3p performance for colorectal cancer detection, increasing its AUC from 0.72 to 0.85 in 48 patients versus 48 controls. Additionally, SE-SPTM-PCR also restores utility to two abandoned biomarkers: For HCMV reactivation monitoring in 32 DNA positive and 32 DNA negative hematopoietic stem cell transplant recipients, hcmv-miR-UL22A-5p achieves an AUC of 0.95. For nasopharyngeal carcinoma, ebv-miR-BART3-3p reaches a perfect AUC of 1.0 in 40 patients and 40 controls. This platform provides a robust tool for miRNA-based liquid biopsy, offering enhanced diagnostic accuracy to support early disease detection and personalized treatment strategies.

MicroRNAs (miRNAs) are short (~22 nucleotides) non-coding RNAs that post-transcriptionally regulate gene expression and are implicated in a wide range of diseases, from cancer to viral infections[1-7]. Their stable presence in biofluids, conferred by association with ribonucleoprotein complexes or extracellular vesicles, makes them promising biomarkers for liquid biopsy[8-10]. However, their clinical translation is hindered by inherent challenges, including their short length, high sequence homology, ultralow abundance in biofluids, and

[1]State Key Laboratory of Vaccines for Infectious Diseases, Xiang An Biomedicine Laboratory, School of Public Health, Xiamen University, Xiamen, Fujian, China. [2]NMPA Key Laboratory for Research and Evaluation of Infectious Disease Diagnostic Technology, School of Public Health, Xiamen University, Xiamen, Fujian, China. [3]The Second Department of Hematology, National Regional Medical Center, Binhai Campus of the First Affiliated Hospital, Fujian Medical University, Fuzhou, China. [4]The Second Department of Hematology, The First Affiliated Hospital, Fujian Medical University, Fuzhou, China. [5]Institute of Precision Medicine, Fujian Medical University, Fuzhou, China. [6]Department of Hematology, Fujian Institute of Hematology, Fujian Provincial Key Laboratory of Hematology, Fujian Medical University Union Hospital, Fuzhou, China. [7]Sun Yat-sen University Cancer Center, State Key Laboratory of Oncology in South China, Collaborative Innovation Center for Cancer Medicine, Guangdong Key Laboratory of Nasopharyngeal Carcinoma Diagnosis and Therapy, Guangzhou, Guangdong Province, China. [8]Department of Nasopharyngeal Carcinoma, Sun Yat-sen University Cancer Center, Guangzhou, Guangdong Province, China. [9]Xiamen Haicang Hospital, Xiamen, Fujian, China. [10]The Second Affiliated Hospital, Fujian Medical University, Quanzhou, China. [11]These authors contributed equally: Zecheng Zhong, Weida Huang, Jinhua Ren, Li Yuan, Qiurong Zhong, Yi Ge. ✉e-mail: szhe@xmu.edu.cn; tanglq@sysucc.org.cn; sxge@xmu.edu.cn; nsxia@xmu.edu.cn; yang.hopeting@gmail.com; zhangshiyin@xmu.edu.cn

susceptibility to matrix interference[11–15]. Furthermore, conventional miRNA detection methods often labor-intensive and lack the sensitivity or specificity needed for robust clinical implementation. To address these limitations, there is an urgent need for convenient, high-performance bioassays capable of accurately quantifying disease associated miRNAs in complex biofluids. Such advancements could unlock the full clinical potential of miRNAs for early disease detection, personalized therapy selection, and real-time treatment monitoring.

Currently, the application of miRNA detection technologies, including northern blotting, hybridization-based microarrays, and sensor assays[16–18], is limited in clinical practice for disease diagnosis due to challenges related to low sensitivity, high costs and complex operations. While reverse transcription-quantitative polymerase chain reaction (RT-qPCR) remains the gold standard for miRNA detection[19], its application to short miRNAs (as opposed to long RNAs like mRNA) requires specialized reverse transcription strategies to enable PCR amplification. Two commonly used strategies for this purpose are polyadenylation and stem-loop RT detection[15,20,21]. The former is notably characterized by poor specificity, whereas the latter enhances the specificity by using stem-loop primers during reverse transcription and fluorescent probes in PCR. This method has been widely adopted in commercial kits and clinically approved assays[22,23]. However, stem-loop RT-qPCR still fails to achieve the specificity of conventional RT-qPCR used for long RNAs such as mRNA. Conventional mRNA RT-qPCR achieves amplification specificity through three distinct sequences elements: the upstream primer, downstream primer, and probe. In contrast, stem-loop RT-qPCR for miRNA lacks sequence specificity in both the downstream primer and probe, relying solely on the stem-loop primer and the upstream primer sequence for specificity. Studies have demonstrated that existing stem-loop RT-qPCR assays can yield strong non-specific amplification, even in template-free water controls[24,25]. This fundamental limitation poses significant challenges for clinical miRNA detection. The compromised specificity in current miRNA detection methods may lead to false positive results and inaccurate quantification in clinical samples. Furthermore, it may indirectly reduce detection sensitivity, potentially obscuring the diagnostic value of low abundance miRNA biomarkers. These limitations highlight the urgent need to develop novel miRNA quantification methods with improved specificity and sensitivity for clinical diagnostics.

The efficiency and reproducibility of miRNA extraction and purification are critical determinants of RT-qPCR accuracy, particularly for low abundance miRNAs in biofluids. Current extraction methods, including phenol-chloroform, guanidinium thiocyanate-based, and column-based techniques[26–28]. However, these methods face challenges such as low enrichment efficiency, the presence of residual inhibitors, poor reproducibility, and the loss of trace miRNAs. Moreover, these methods co-purify total RNA, exposing target miRNAs to interference from homologous miRNA family members and abundant background RNAs. For instance, similar sequences may be present at 1000-fold higher concentrations than the target miRNA, leading to cross reactivity and false quantification. Additionally, the reliance on manual processing introduces operator dependent variability and limits scalability for high throughput clinical applications. Therefore, it is essential to develop a miRNA extraction method that is highly efficient, automated, reproducible, and resistant to interference from similar miRNAs and background RNAs.

To address the unmet needs in clinical miRNA detection, we develop a Selective Enrichment miRNA automated extraction and Specific-Probe Terminal-Mediated RT-qPCR (SE-SPTM-PCR) platform for the precise, sensitive, and specific detection of miRNAs in biofluid samples. Compared to existing miRNA quantification methods, the SE-SPTM-PCR platform offers four notable advantages. First, we establish a solid-phase magnetic bead surface DNA-guided miRNA selective enrichment strategy to accurately identify and enrich disease-sensitive miRNAs, which represent a small yet crucial subset of total RNA. This approach reduces potential background RNA interference and enhances the accuracy of miRNA based liquid biopsies. Specifically, DNA probe modified with miRNA strand-specific nucleotide analogs are immobilized on the surface of solid-phase magnetic beads to create unique identity recognition barcodes, enabling selective recognition and controllable binding of miRNAs, thereby achieving selective enrichment. Second, primers are designed with Tm values closely aligned to the annealing temperature to enhance sensitivity in recognizing miRNA variant bases. This approach enables the accurate and sensitive identification of single or multiple base variations across entire miRNA sequences with strand specificity, further improving the specificity of disease-related miRNA detection. Third, the 3' end of the fluorescent probe is engineered with a unique miRNA binding sequence, providing for recognition specificity during the amplification process and enabling sensitive identification of miRNA base variations. When combined with the primers, this design facilitates the sensitive and accurate identification of base variations across the entire miRNA site. This strategy not only enhances the specificity of the miRNA detection system itself but also improves discriminatory power for disease-related miRNAs. Fourth, we integrate multiple processes into SE-SPTM-PCR, including automated miRNA extraction, reverse transcription of all miRNA templates following purification and elution, and PCR amplification utilizing the full cDNA input. This integration creates a highly sensitive miRNA assay platform that combines extraction, reverse transcription, amplification, and detection. SE-SPTM-PCR demonstrates exceptional analytical performance for liquid biopsy applications, significantly enhancing analytical specificity and increasing analytical sensitivity by 100-fold compared to the traditional stem-loop RT-qPCR method. Using clinical samples for evaluation, we significantly improve the diagnostic performance of hsa-miR-92a-3p in colorectal cancer. Additionally, we rediscover the diagnostic significance of hcmv-miR-UL22A-5p for HCMV reactivation in patients undergoing hematopoietic stem cell transplantation as well as the clinical value of ebv-miR-BART-3-3p in diagnosing nasopharyngeal carcinoma. We envision that SE-SPTM-PCR provides a promising tool for advancing miRNA detection, accelerating clinical translation of miRNA biomarkers, and enhancing the clinical applicability of miRNA-based liquid biopsies.

## Results

### Rational design of SE-SPTM-PCR system

Figure 1 illustrates the schematic diagram of miRNA detection in human plasma or serum samples following selective enrichment and SPTM-PCR technology. In this process, streptavidin magnetic beads conjugated with 3' biotin-labeled probe DNA (DNA-SAMB) were employed to capture target miRNAs. To enhance hybridization specificity and affinity, locked nucleic acid (LNA) was incorporated into the DNA probes, which strengthens the non-covalent interactions between nitrogenous bases and hydrogen bonds (Fig. 1a). To mitigate potential variability in miRNA capture efficiency due to fluctuating miRNA quantities, a molar excess of DNA-SAMB was introduced into lysed plasma or serum samples. Through sequence-specific complementary pairing, the DNA capture probes selectively bound target miRNAs within the complex biological matrix. The DNA/miRNA-beads complexes were separated magnetically, while unhybridized sequences were washed away. Subsequently, the purified target miRNAs were eluted under high temperature conditions (Fig. 1b). Notably, the selective enrichment process could not fully discriminate between the target miRNA and closely related sequences differing by only a few nucleotides, particularly when nucleotide mismatches occurred at the 3' end of the miRNA. Therefore, both target miRNAs and their sequence homologs were subjected to reverse transcription. Crucially, sequence variants with 3'-end mismatches exhibited reduced thermodynamic stability during stem-loop primer binding, thereby

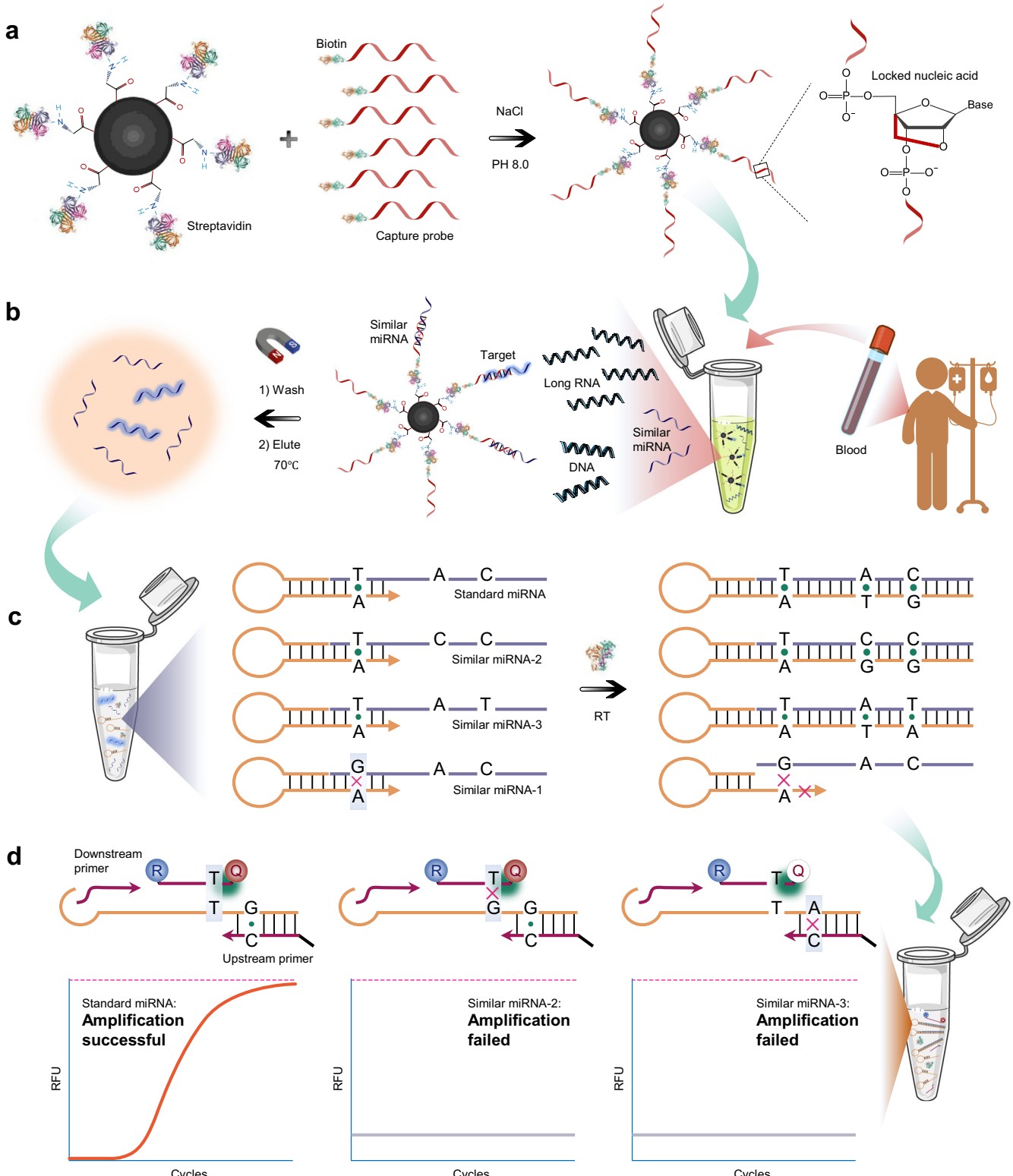

**Fig. 1 | Schematic of the SE-SPTM-PCR system for miRNA detection in biological fluid samples. a** Functionalized miRNA capture carriers are prepared by reacting streptavidin (PDB: 1DF8) magnetic beads with biotin (PDB: 1WPY) and locked nucleic acid-modified capture probes. **b** The capture probe-labeled magnetic beads selectively enrich miRNA in plasma or serum, purifying target miRNA molecules along with some similar miRNAs or RNAs. The blood donor image designed by Freepik. **c** During reverse transcription, standard miRNA and similar miRNAs with several different sites outside the stem-loop primer specific pairing region are reverse transcribed. In contrast, similar miRNAs within this region are fail to be recognized and transcribed. Reverse transcriptase (PDB: 3KLE). **d** In SPTM-PCR, target miRNA is amplified normally. However, mismatches between variant bases in similar miRNAs and the specific upstream primer or fluorescent probe reduce thermodynamic binding capacity, preventing non-specific amplification signals. Free illustration materials are adapted from Bioicons. RFU relative fluorescence units. R Reporter, Q Quencher.

preventing cDNA extension. In contrast, variants with mismatches in the non-complementary 5′ region successfully underwent reverse transcription (Fig. 1c). All cDNA products were subsequently amplified in the qPCR system. The target miRNA-derived cDNAs, being fully complementary to both the fluorescent probe and upstream primer, demonstrated efficient amplification. Conversely, sequence homologs containing 5′-end mismatches failed to amplify due to impaired probe or primer binding stability. Of particular significance, the fluorescent probe incorporated sequence-specific recognition elements at its 3′ terminus, substantially enhancing detection specificity (Fig. 1d). The detection system features a triple-specificity design. First, capture probes to remove background nucleic acids. Second, stem-loop primers to distinguish miRNAs with 3′-end sequence variations. Third, fluorescent probe paired with upstream primers to differentiate miRNAs with 5′-end homology. This tandem strategy ensures specificity throughout the entire detection process.

### The miRNA selective enrichment method effectively removes background nucleic acids

To address background nucleic acid contamination in traditional miRNA extraction methods, we developed a selective miRNA enrichment approach. The working principle involves the conjugation of magnetic beads with capture probes, miRNA hybridization, removal of background nucleic acids, and elution of target miRNA (Supplementary Fig. S1a–d). We systematically optimized critical parameters, including magnetic bead type and density, capture probe design (type, structure, length, and affinity), lysis buffer composition and concentration, and elution conditions (Supplementary Figs. S2–7). This optimization culminated in an automated miRNA selective enrichment protocol (Supplementary Tables S1 and S2).

To evaluate recovery efficiency, we applied this method to synthetic miRNA standards. The results demonstrated an average recovery rate exceeded 50% across all tested miRNAs (Supplementary Fig. S8a, b). Subsequently, we compared the extraction performance of the selective enrichment method with that of traditional commercial methods. First, analysis of hsa-miR-122-5p extraction from 30 clinical plasma samples of healthy humans revealed that the selective enrichment method achieved higher yields in 28 samples, with only 2 samples showing comparable yields to the traditional method (Supplementary Fig. S9). Second, the selective enrichment method exhibited superior intra- and inter-batch reproducibility, showing coefficients of variation ≤1%, significantly lower than those of traditional methods (Supplementary Fig. S10, 11). Third, parallel testing of samples extracted by both methods revealed amplification inhibition in traditional methods but normal amplification with selective enrichment (Supplementary Fig. S12), demonstrating the latter's ability to obtain higher purity extracts. Fourth, the selective enrichment method consistently outperformed traditional methods in extraction yield across routine clinical samples, hemolyzed samples, and lipemic samples (Supplementary Fig. S13), demonstrating broad applicability. Additional advantages included reduced processing time, operational convenience, and cost-effectiveness (Supplementary Table S3).

To investigate the method's capacity to address background nucleic acid contamination, we employed a stepwise strategy from simple to complex systems. We first examined common contaminants in biological fluids, including double-stranded DNA, single-stranded DNA, long chain RNA, and non-target miRNAs with significant sequence variations. The selective enrichment method recovered 63% of target miRNA but <8% of background nucleic acids (Fig. 2a). To assess specificity of identifying sequence similar miRNAs, we synthesized hsa-miR-122-5p mutants with 1–3 base differences (Supplementary Table S4). The results demonstrated that the detection Cq values of the target miRNA before and after recovery did not change significantly, while the Cq values of the mutants shifted to a later stage

after recovery, with the extent of the shift increasing as the number of mutant bases increased (Fig. 2b). Finally, we spiked known miRNA standards into lysed plasma with complex backgrounds. Traditional silica column extraction caused strong non-specific amplification, obscuring positive and negative group differentiation and reducing sensitivity (Fig. 2c). In contrast, selective enrichment enabled clear discrimination between groups (Fig. 2d). Collectively, these results demonstrate that the selective enrichment method effectively eliminates background nucleic acids and enhances the specificity.

### SPTM-PCR demonstrates excellent detection specificity

Although stem-loop RT-qPCR is known to exhibit inherent non-specific amplification[24,25], systematic large-scale evaluations have been lacking. Following established primer and probe design rules[15], we developed stem-loop RT-qPCR assays for all 26 HCMV and 44 EBV miRNAs to assess specificity (Supplementary Table S5, 6). Strikingly, approximately 33% of HCMV and 25% of EBV assays exhibited non-specific amplification even in no-template water controls (Supplementary Fig. S14a, b). Mechanistic investigation through primers and probe deconstruction revealed that non-specific amplification required co-presence of both stem-loop and upstream primers (Supplementary Fig. S15a–e). Dilution experiments further demonstrated concentration dependent reduction of non-specific signals in these primers, whereas downstream primers and probes showed no such effects (Supplementary Fig. S16a–d). Next-generation sequencing of amplification products identified sequence mismatches between non-specific and positive controls (Supplementary Fig. S17a–c), consistent with primer-primer interactions between the stem-loop primer and upstream primer 3′ regions (Supplementary Fig. S17d). Conventional stem-loop RT-qPCR cannot discriminate these artifacts due to probe and stem-loop primer sequence overlap (Fig. 3a). To enhance specificity, we engineered the 3′ end of probe for miRNA-specific recognition (Fig. 3b). Specifically, truncation of the 3′ ends of both upstream and stem-loop primers resulted in sensitivity to 3′ single-base deletions while maintaining high amplification efficiency when primer Tm approached actual annealing temperatures (Supplementary Fig. S18 a, b, d, e). This truncation strategy enabled miRNA-specific sequence incorporation at the 3′ end of probe. Additionally, 5′ end truncation of the probe achieved sensitivity to deletion mutations when probe Tm matched annealing conditions (Supplementary Fig. S18c, f). We designated this method as Specific Probe Terminus-Mediated RT-qPCR (SPTM-PCR).

Compared to traditional stem-loop RT-qPCR, SPTM-PCR demonstrates significantly enhanced specificity, attributable to its optimized probe design (Fig. 3c, d). To systematically evaluate this advantage, we validated the SPTM-PCR system using HCMV-encoded hcmv-miR-UL22A-5p under progressively increasing template complexity. First, in negative control testing with 46 template-free water samples, the SPTM-PCR system exhibited no detectable non-specific amplification (Fig. 3e), whereas the commercially available stem-loop RT-qPCR kit showed false-positive signals in 33.3% (4/12) of blank water samples (Fig. 3f). To further evaluate detection performance in biologically complex matrices, we analyzed nucleic acid extracts from 20 mouse and 20 healthy human plasma samples using silica column purification. Remarkably, while the reference method demonstrated near complete non-specific amplification across these complex biological samples, the SPTM-PCR system maintained absolute specificity without any spurious signals (Supplementary Fig. S19a–f). These results collectively indicate that SPTM-PCR not only achieves intrinsic analytical specificity but also retains exceptional discrimination capability when detecting targets in challenging biological matrices. Furthermore, the method demonstrated excellent reproducibility metrics with minimal intra- and inter-batch variability (Supplementary Fig. S20a, b).

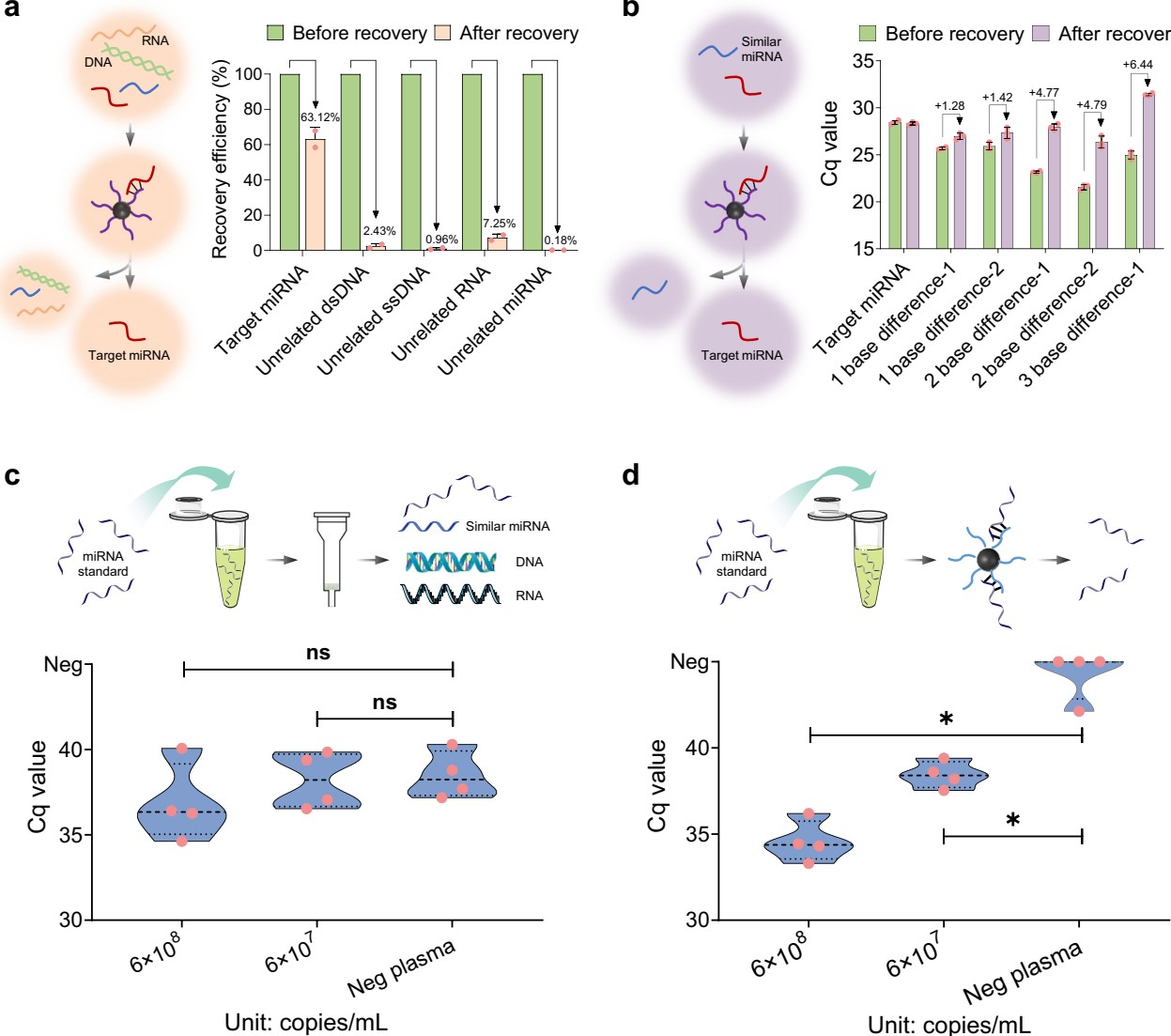

**Fig. 2 | Validation of selective enrichment method performance in addressing background nucleic acid contamination. a** The selective enrichment method extracts target miRNA, double-stranded DNA, single-stranded DNA, non-relevant long chain RNA, and unrelated miRNA (with known concentrations), and the extraction efficiency is calculated based on recovery content ($n$ = 2 independent experiments). **b** The removal performance of the selective enrichment method for similar miRNAs with 1–3 base differences is validated ($n$ = 3 independent samples),

the data are displayed as mean ± s.d. A known concentration of miRNA standard is mixed with lysed plasma, and detection is performed after extraction using (**c**) a silica column membrane and (**d**) the selective enrichment method. Free illustration materials are adapted from Bioicons. Two-tailed Mann-Whitney U test. *$P$ < 0.05. Cq Quantification cycle. The ns means not significant. The neg means negative. Source data are provided as a Source Data file.

## SE-SPTM-PCR enables high specificity and sensitivity in miRNA detection

The ability to detect miRNA with high sensitivity is critical for early identification of minor expression changes, which can advance disease diagnosis, progression monitoring, and personalized medicine development. We developed a high-sensitivity detection system, SE-SPTM-PCR, by combining selective enrichment with specific probe terminal mediated RT-qPCR (Fig. 4a). Unlike conventional miRNA detection methods, this system maximizes the utilization of enriched miRNA and cDNA products while capitalizing on SE's high enrichment efficiency and SPTM-PCR's target specificity (Supplementary Fig. S21a, b). As a result, the system achieves both high sensitivity and specificity in miRNA detection. When detecting artificially synthesized miRNA at 2-fold gradient dilutions, SE-SPTM-PCR showed a sensitivity of up to $10^3$ copies/mL (Supplementary Fig. S22a, b).

To evaluate the performance of SE-SPTM-PCR in real world samples, we compared it with traditional miRNA detection method. For

specificity assessment, we tested 20 healthy human plasma samples (confirmed HCMV DNA-negative) using HCMV virus miRNA reagents. SE-SPTM-PCR showed no nonspecific amplification (Fig. 4b), whereas traditional methods produced nonspecific signals in nearly all samples (Fig. 4c). This superior specificity stems from the intrinsic design of SE-SPTM-PCR reagents, which precisely target the miRNA of interest while minimizing background nucleic acid interference. To compare sensitivity, we analyzed 10-fold serially diluted HCMV-infected cell culture supernatant mixed with healthy plasma. SE-SPTM-PCR exhibited 100-fold higher sensitivity than traditional method (Fig. 4d–f), surpassing the theoretical 40-fold improvement predicted from template utilization efficiency alone (Supplementary Fig. S21a, b). This discrepancy likely arises from compounded effects of background interference reduction and system-specific optimization. Based on prior reports and our empirical data[15,29], we estimate the practical sensitivity enhancement of SE-SPTM-PCR to range between 40- and 10,000-fold across different miRNA reagents.

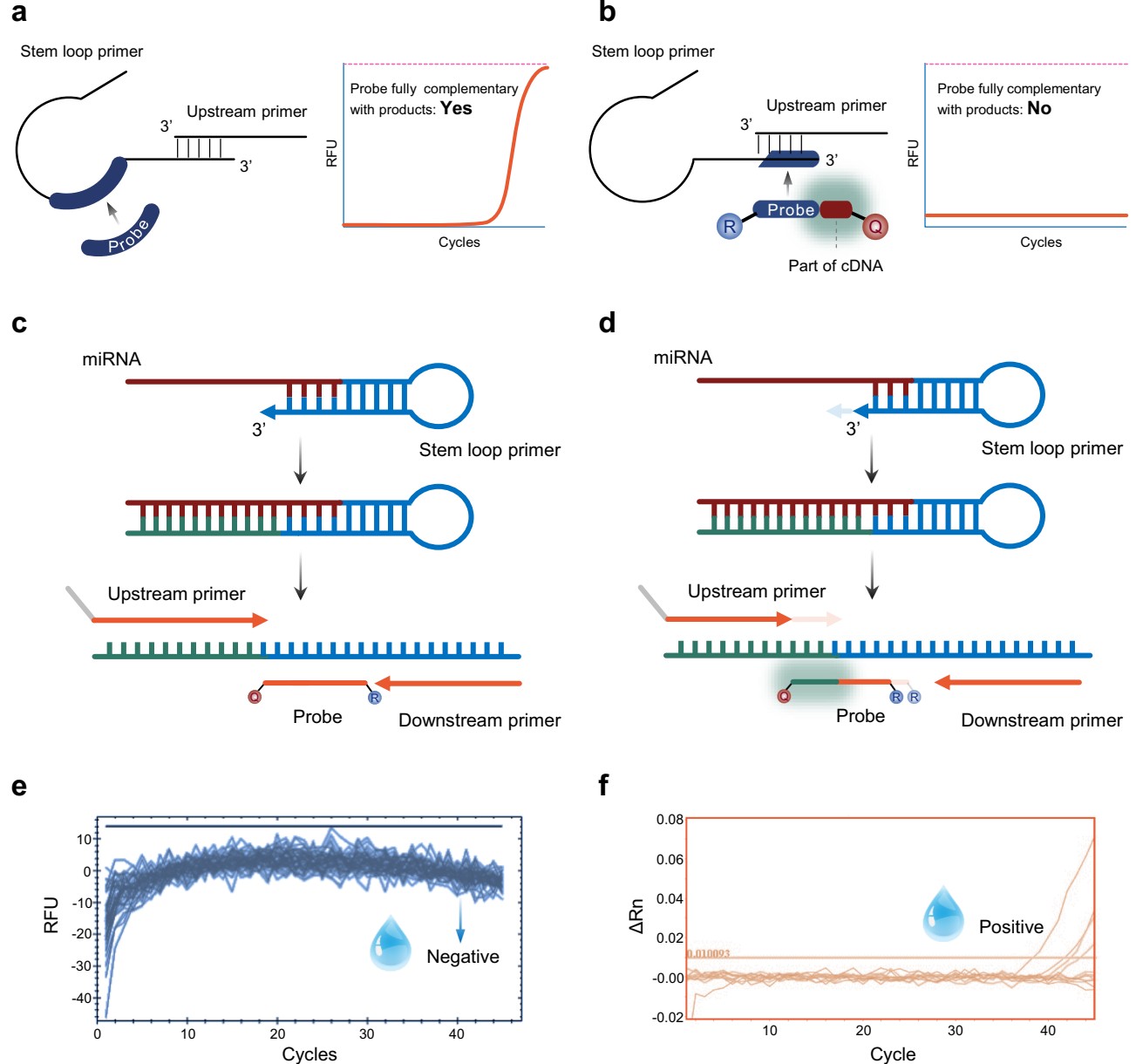

**Fig. 3 | Principle of the SPTM-PCR method and validation of detection specificity. a** The traditional stem-loop primer RT-qPCR method's probe lacks miRNA recognition specificity during the qPCR process, making it susceptible to non-specific amplification mediated by the upstream primer and stem-loop primer, which can easily lead to false-positive signals. Free illustration materials are adapted from Bioicons. **b** By incorporating an miRNA specific recognition segment into the TaqMan probe, non-specific amplification products can be effectively identified.

Free illustration materials are adapted from Bioicons. **c** Schematic diagram of miRNA detection using the conventional stem-loop primer RT-qPCR method. **d** Schematic diagram of miRNA detection using the SPTM-PCR method. **e** SPTM-PCR detection of no template water sample (n = 46). The water droplet image designed by Freepik. **f** Commercial ABI stem-loop primer RT-qPCR detection of no template water sample (n = 12). The water droplet image designed by Freepik. RFU relative fluorescence units. ΔRn delta normalized reporter.

## SE-SPTM-PCR enhances the diagnostic performance of human miRNAs in colorectal cancer

Human peripheral blood contains abundant endogenous miRNAs with demonstrated diagnostic potential for malignancies[5]. Using colorectal cancer (CRC) as a model system, we focused on previously reported human miRNAs with CRC diagnostic value (Fig. 5a), including hsa-miR-92a-3p[30–32], hsa-miR-19a-3p[32], hsa-miR-320a, and hsa-miR-423-5p[33]. We selected these targets to evaluate the diagnostic applicability of SE-SPTM-PCR (Supplementary Table S7) for human miRNA detection in cancer.

Age and gender matched serum samples from CRC patients and healthy controls were analyzed for these four human miRNAs expression (Supplementary Table S8). We established two

conventional detection workflows using commercially available extraction kits and qPCR reagents, which were directly compared with our SE-SPTM-PCR protocol in parallel sample analyses (Supplementary Fig. S23). Analysis of Cq values revealed differential diagnostic performance across methods. In conventional detection scheme 1, the average expression level of hsa-miR-92a-3p in CRC patients was significantly higher than in healthy individuals (p < 0.001), but this method showed the lowest diagnostic performance, with ROC curve analysis yielding an area under the curve (AUC) of 0.72 (Fig. 5b). In conventional scheme 2, hsa-miR-92a-3p expression was significantly higher in CRC patients (p < 0.0001), achieving an AUC of 0.80 (Fig. 5c). Strikingly, SE-SPTM-PCR demonstrated superior diagnostic performance, the difference in

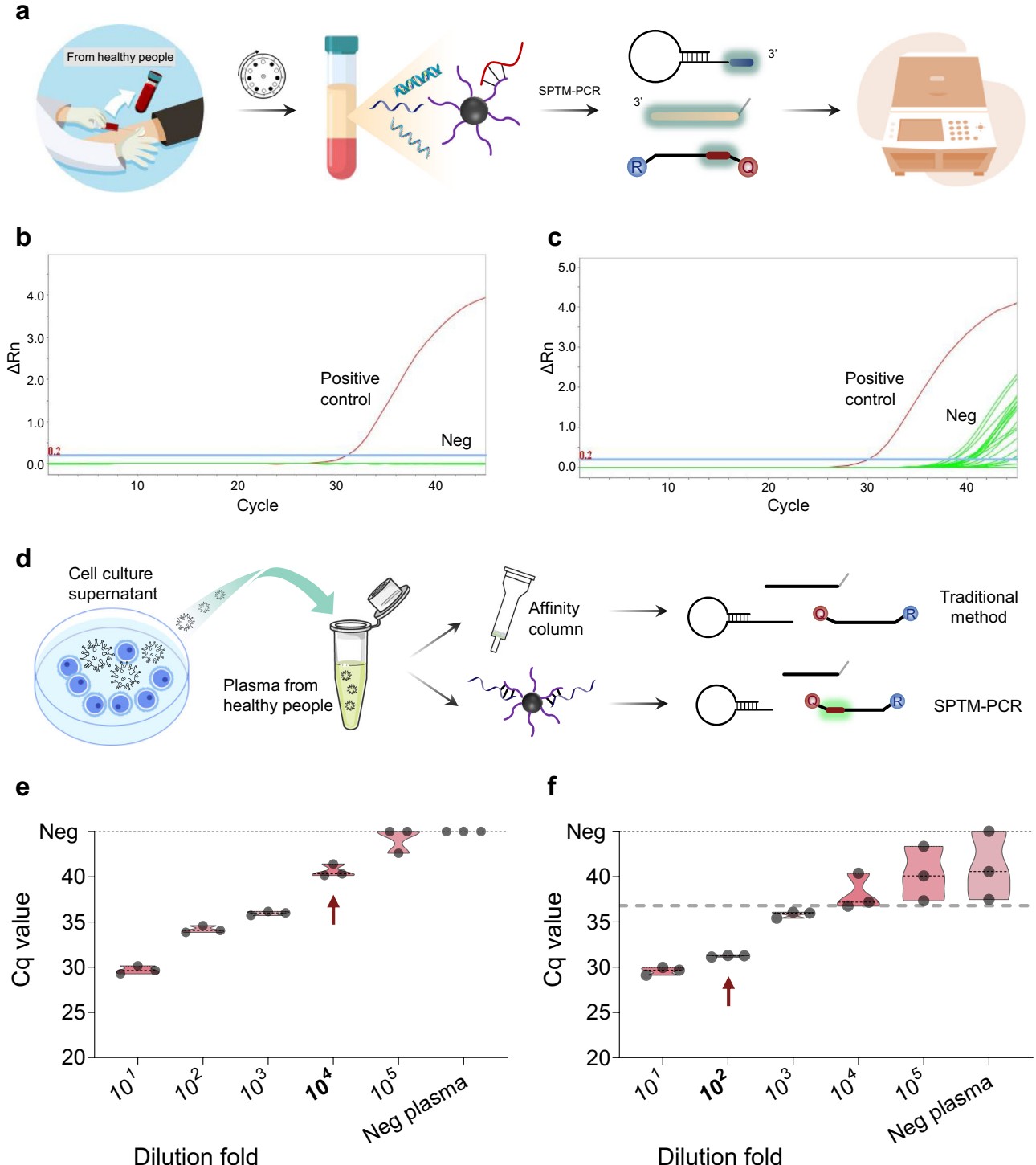

**Fig. 4 | Development and analytical performance of the SE-SPTM-PCR high sensitivity detection system. a** Workflow of the SE-SPTM-PCR high sensitivity detection system. Free illustration materials are adapted from Bioicons. The blood collection process, blood collection tube and fluorescence PCR instrument images designed by Freepik. **b** Detection of plasma samples from healthy individuals without HCMV infection using the SE-SPTM-PCR protocol (*n* = 20). **c** Detection of plasma samples from healthy individuals without HCMV infection using the conventional protocol (*n* = 20). **d** Schematic of the SE-SPTM-PCR and conventional protocols for detecting serially diluted HCMV cell culture supernatant spiked into healthy human plasma. Free illustration materials are adapted from Bioicons. The cell culture dish image designed by Freepik. **e** Limit of detection assessment for the SE-SPTM-PCR system in plasma. **f** Limit of detection assessment for the conventional protocol in plasma. The neg means negative. ΔRn delta normalized reporter, Cq Quantification cycle. Source data are provided as a Source Data file.

hsa-miR-92a-3p expression between CRC patients and healthy controls was more pronounced, with an AUC of 0.85 (Fig. 5d). The validation results for hsa-miR-19a-3p further confirmed the better diagnostic performance of the SE-SPTM-PCR-based approach compared to the conventional method (Supplementary Fig. S24a–c). For hsa-miR-320a and hsa-miR-423-5p, the SE-SPTM-PCR assay performed equivalently to or better than the conventional assay (Supplementary Fig. S24d–i). These results indicate that SE-SPTM-PCR could enhance the diagnostic accuracy of human miRNAs for CRC detection compared to conventional methods.

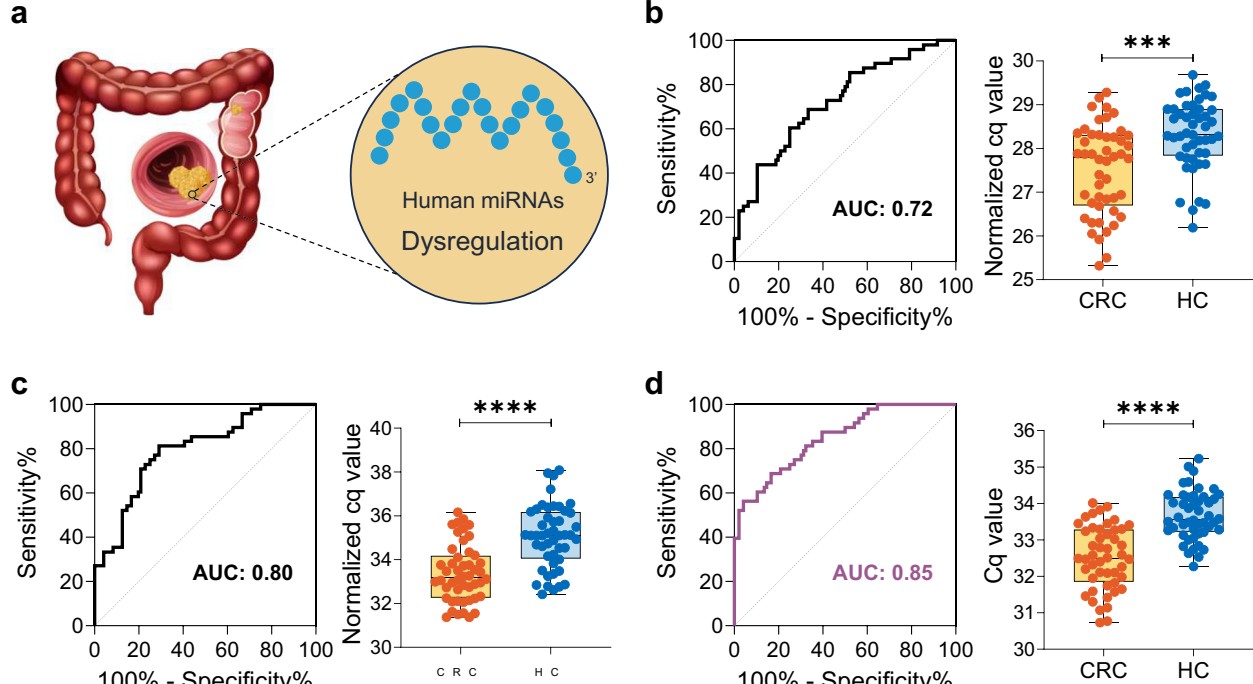

**Fig. 5 | Diagnostic performance comparison of SE-SPTM-PCR versus conventional miRNA detection methods for CRC diagnosis using hsa-miR-92a-3p.**
**a** Schematic illustrating the diagnostic potential of dysregulated human miRNAs in CRC. The image of schematic diagram of colorectal cancer designed by Freepik. **b** Diagnostic performance of conventional scheme 1 (TIANGEN column extraction combined with stem-loop primer RT-qPCR). **c** Diagnostic performance of conventional scheme 2 (QIAGEN column extraction combined with stem-loop primer RT-qPCR). **d** Diagnostic accuracy of the SE-SPTM-PCR based detection system. The boxes indicate the interquartile range (IQR) of data between 75% (Q3) and 25% (Q1). The bars below and above each box indicate the data in Q1−1.5 × IQR and Q3 + 1.5 × IQR, respectively. Two-tailed Mann-Whitney U test. CRC colorectal cancer patients, $n = 48$ biologically independent samples; HC healthy controls, $n = 48$ biologically independent samples. AUC area under the curve. ***$P < 0.001$, ****$P < 0.0001$. Source data are provided as a Source Data file.

## SE-SPTM-PCR confers diagnostic value to HCMV miRNAs for HCMV infection in hematopoietic stem cell transplantation

HCMV miRNAs have been widely investigated as potential biomarkers for HCMV infection in various diseases[34–37]. However, recent evidence suggests that certain HCMV miRNAs (including hcmv-miR-UL22A-5p) lack diagnostic utility for detecting HCMV reactivation in hematopoietic stem cell transplantation (HSCT) recipients[38]. Here, we evaluated whether SE-SPTM-PCR (Supplementary Table S7) could enhance the diagnostic performance of HCMV miRNAs.

HCMV infection or reactivation in HSCT recipients is associated with severe complications, including HCMV disease, graft-versus-host disease, and increased mortality[39–41] (Fig. 6a). Early detection through blood viral marker monitoring followed by preemptive therapy constitutes a critical preventive strategy. HCMV DNA has replaced antigen detection as the preferred reactivation marker in HSCT patients due to superior sensitivity[42]. However, HCMV reactivation remains a major clinical challenge[39,43–47], necessitating more sensitive monitoring tools. We analyzed plasma samples from 32 HCMV DNA positive and 32 DNA negative HSCT patients using both SE-SPTM-PCR and conventional miRNA detection. The results showed that conventional method failed to discriminate HCMV DNA positive and negative patients through qualitative detection of hcmv-miR-US25-1-5p (Fig. 6b). In stark contrast, SE-SPTM-PCR enabled significant differentiation between these groups (Fig. 6c), achieving a diagnostic AUC of 0.96 (Fig. 6d). This performance advantage extended to four additional HCMV miRNAs (Supplementary Fig. S25a–l). SE-SPTM-PCR also significantly outperformed miRNA deep sequencing in resolving slight differences, as demonstrated using 2-fold gradient dilutions of simulated viral infection samples (Supplementary Fig. S26a, b). Quantitative correlation analysis revealed a strong positive association between HCMV DNA and miRNA levels (Pearson's $r = 0.82$, $p < 2.2e-16$; Fig. 6e). Notably,

HCMV miRNA viral loads exceeded DNA levels (Fig. 6f), suggesting that miRNA detection via SE-SPTM-PCR may offer superior sensitivity for early diagnosis of HCMV infection or reactivation.

## SE-SPTM-PCR improves the diagnostic performance of EBV miRNAs in nasopharyngeal carcinoma

EBV miRNA is closely associated with the development and progression of nasopharyngeal carcinoma (NPC). Numerous studies have reported the potential value of circulating EBV miRNA in NPC diagnosis, prognosis, treatment efficacy evaluation, and disease monitoring[48–52]. However, to date, none of these miRNAs has been successfully translated into clinical applications. In this study, we hypothesized that SE-SPTM-PCR could enhance the diagnostic performance of existing EBV miRNA biomarkers for NPC. To validate this, we selected three previously reported targets: the diagnostic biomarkers ebv-miR-BART7-3p and ebv-miR-BART2-5p[48,49], along with non-diagnostic ebv-miR-BART3-3p for further analysis[48].

Plasma samples from NPC patients and healthy controls were collected (Fig. 7a) and analyzed using both SE-SPTM-PCR (Supplementary Table S7) and conventional miRNA detection methods. Conventional method showed statistically significant differences in EBV miRNA levels between groups, but discriminatory capacity was limited (Fig. 7b, d, f). In contrast, SE-SPTM-PCR analysis of the same samples demonstrated clear discrimination between NPC patients and healthy controls, with no non-specific amplification observed in healthy individuals or no-template water controls (Fig. 7c, e, g). Notably, ebv-miR-BART3-3p, previously regarded as non-diagnostic for NPC, exhibited excellent discriminatory capability when detected by SE-SPTM-PCR (Fig. 7d). All three EBV miRNAs achieved near-perfect diagnostic accuracy (AUC = 1.0), markedly superior to conventional detection (Fig. 7h, i). Although the enrolled NPC patients were predominantly at

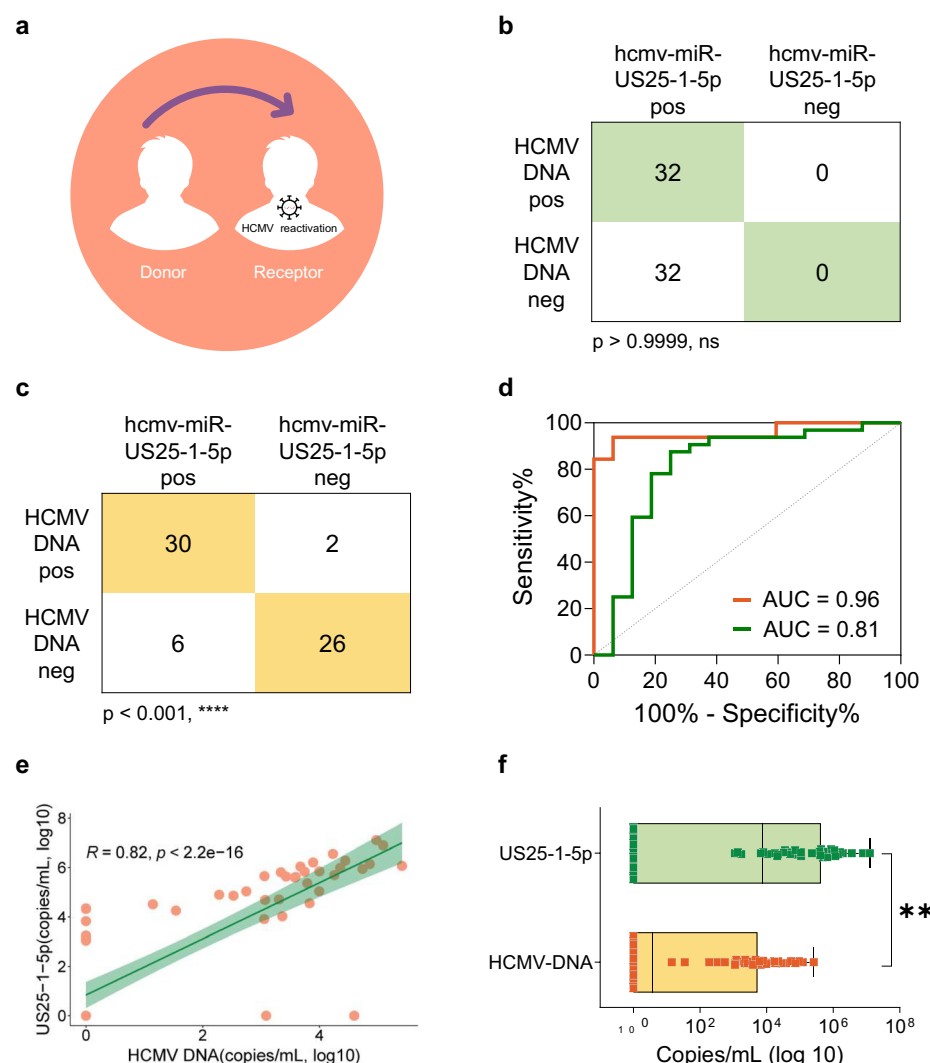

**Fig. 6 | Diagnostic performance comparison of SE-SPTM-PCR and conventional miRNA detection systems for monitoring HCMV reactivation in HSCT recipients. a** Schematic of HCMV reactivation post HSCT. **b** Detection in HCMV DNA positive and negative HSCT patients using conventional RT-qPCR. Chi-square tests. **c** Detection in HCMV DNA positive and negative HSCT patients using SE-SPTM-PCR. Chi-square tests. **d** ROC curve analysis of SE-SPTM-PCR and conventional RT-qPCR detection performance. Orange line, SE-SPTM-PCR. Green line, conventional RT-qPCR. **e** Correlation between HCMV DNA load and miRNA levels. Pearson's $r = 0.82$. **f** Viral load comparison between DNA and miRNA detection methods. The boxes indicate the interquartile range (IQR) of data between 75% (Q3) and 25% (Q1). The bars below and above each box indicate the data in Q1 − 1.5 × IQR and Q3 + 1.5 × IQR, respectively. Two-tailed Mann-Whitney U test. The pos means positive. The neg means negative. The ns means not significant. CI confidence interval. **$P < 0.01$. Source data are provided as a Source Data file.

advanced stages (Supplementary Table S9), the high sensitivity of SE-SPTM-PCR suggests its potential utility for improving early-stage NPC detection in future clinical applications.

## Discussion

MiRNAs hold significant potential as biomarkers for liquid biopsy due to their easy accessibility in all bodily fluids and their stability within these fluids, as they are either protein-coated or encapsulated in vesicles, which protects them from RNase degradation[8–10]. Over the past decade, numerous reports have emerged regarding miRNAs serving as liquid biopsy biomarkers for various diseases[53,54]. However, the clinical application of miRNA-based liquid biopsy lags far behind laboratory research, a discrepancy primarily attributed to the complex contamination of nucleic acids in biological fluids and the specificity challenges of miRNA detection. Although traditional separation techniques can classify miRNAs based on characteristics such as size, charge, and density[26–28], even the most thorough separation

procedures may still yield heterogeneous RNAs or similar miRNAs. In biological fluids, the proportion of miRNAs is low, and their short length, along with high sequence similarity within families, undoubtedly renders miRNA detection results susceptible to interference from complex backgrounds. Although the stem-loop RT-qPCR method can achieve specific miRNA detection by designing primers for differential base sites, which has proven effective in distinguishing family miRNAs[21,55,56], it still suffers from poor specificity[24,25], thereby increasing the uncertainty of detection results. Furthermore, this method has limited capability in recognizing similar miRNAs or RNAs that lack differential bases at the 3' end. Therefore, our work aims to establish a platform with broad adaptability that minimizes interference from background nucleic acids and enhances the specificity, sensitivity, and accuracy of miRNA detection.

Regarding miRNA isolation, traditional techniques primarily based on Trizol, silica column membranes, or non-specific magnetic bead technology may be influenced by coexisting components of

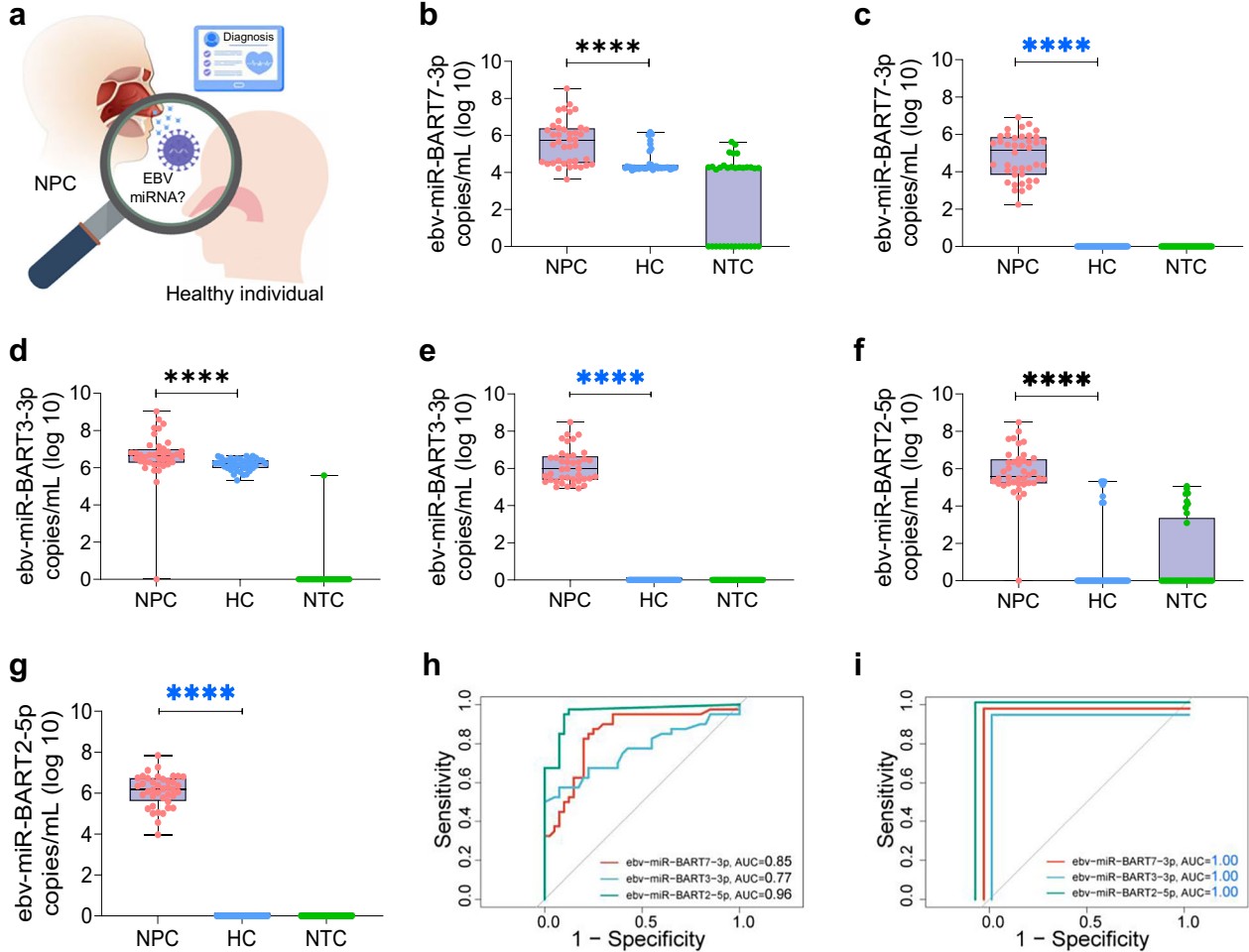

**Fig. 7 | Diagnostic performance evaluation of SE-SPTM-PCR versus conventional RT-qPCR for NPC detection using EBV miRNAs. a** Schematic illustrating EBV miRNA diagnostic potential for NPC. The images of diagnostic schematic of NPC designed by Freepik. **b–g** Clinical validation results: **b** Detection of ebv-miR-BART7-3p by conventional RT-qPCR. **c** Detection of ebv-miR-BART7-3p by SE-SPTM-PCR. **d** Detection of ebv-miR-BART3-3p by conventional RT-qPCR. **e** Detection of ebv-miR-BART3-3p by SE-SPTM-PCR. **f** Detection of ebv-miR-BART2-5p by conventional RT-qPCR. **g** Detection of ebv-miR-BART2-5p by SE-SPTM-PCR.

**h, i** Diagnostic performance of the EBV miRNA system based on conventional RT-qPCR and SE-SPTM-PCR for NPC. NPC nasopharyngeal carcinoma, HC healthy controls, NTC no templates water, AUC area under the curve. (NPC, *n* = 40 biologically independent samples; HC, *n* = 40 biologically independent samples). The boxes indicate the interquartile range (IQR) of data between 75% (Q3) and 25% (Q1). The bars below and above each box indicate the data in Q1 − 1.5 × IQR and Q3 + 1.5 × IQR, respectively. ****P < 0.0001. Source data are provided as a Source Data file.

the target miRNA, such as interference from similar miRNAs, thus lacking disease specificity. In this study, we propose a selective enrichment miRNA automated extraction technique that precisely captures disease related miRNAs in biological fluids. Our method introduces specific capture probes targeting the desired miRNAs, allowing for targeted recognition of complementary DNA and selective capture enrichment. Unlike traditional separation techniques, we utilize the selectivity of selective enrichment to minimize interference from other non-target nucleic acid molecules and similar miRNAs, thereby enabling rapid and accurate identification of disease related miRNAs in biofluid samples. The automated extraction with superparamagnetic beads avoids lengthy separation and purification processes while minimizing non-specific interference from background contamination. Notably, compared to traditional separation techniques, selective enrichment miRNA extraction technology offers higher miRNA yield, better reproducibility, higher purification efficiency, shorter processing time, greater convenience, lower cost, and improved sample applicability. These advantages make selective enrichment miRNA extraction technology highly suitable for the detection of routine clinical samples.

In the quantitative analysis of miRNAs, most existing techniques are susceptible to interference from background nucleic acids or similar miRNAs[13], which may be attributed to the challenges associated with detecting short sequence miRNAs using current technologies. Even the stem-loop RT-qPCR method, regarded as the 'gold standard' for miRNA detection, still faces issues with insufficient specificity in its methodology[24,25]. In this study, we discovered that the 3' end sequence of the stem-loop primer and the 3' end sequence of the upstream primer exhibit partial base complementary pairing, which mediates non-specific extension and amplification reactions. Based on this finding, we proposed a strategy for miRNA detection, termed SPTM-PCR. Compared to the traditional stem-loop RT-qPCR technique, SPTM-PCR offers two notable advantages. First, SPTM-PCR incorporates miRNA-specific bases at the 3' end of the probe, which enhances specificity compared to traditional probes and improves the system's tolerance to non-specific amplification. Second, this technology exhibits strong resistance to interference from co-existing nucleic acids in complex biological matrices. Conventional miRNA detection methods often suffer from significant non-specific amplification due to residual background nucleic acids that persist after silica column membrane extraction. In stark contrast, SPTM-PCR eliminates non-specific

amplification entirely, achieving near-zero background interference. This enhanced performance likely stems from the superior target recognition capability and exceptional detection specificity inherent to SPTM-PCR. Beyond the miRNA model utilized in this study, this method is theoretically applicable to all other types of miRNA detection, paving the way for non-invasive, high precision disease diagnosis and basic research, among other fields, thereby demonstrating its broad application prospects.

The development of ultrasensitive detection methodologies is a critical prerequisite for identifying disease associated miRNA signatures in complex biological matrices. This necessity arises from the characteristically low abundance of circulating miRNA biomarkers, particularly during early disease pathogenesis when timely diagnosis can significantly impact clinical outcomes. This study leverages the advantages of miRNA selective enrichment technology and SPTM-PCR technology, combined with strategies to enhance template and cDNA utilization, to establish a highly sensitive miRNA detection platform (SE-SPTM-PCR). The sensitivity of the SE-SPTM-PCR miRNA platform is 100 times higher than that of traditional miRNA detection platforms. The superiority of this high sensitivity technology primarily stems from several key factors. Firstly, selective enrichment technology boasts higher enrichment and miRNA extraction efficiency, capturing more miRNAs and preventing their loss. Secondly, the SPTM-PCR method exhibits superior detection specificity, which indirectly enhances the sensitivity of miRNA detection. Thirdly, compared to traditional miRNA detection platforms that employ strategies such as template dilution or reduced template input to mitigate the impact of residual extraction inhibitors on qPCR amplification, the selective enrichment method, which more thoroughly removes inhibitors than commercially available silica column membrane extraction, avoids interference with qPCR amplification. Consequently, our platform can utilize the entire extracted and purified miRNA template for reverse transcription. Finally, in the traditional stem-loop RT-qPCR method, to mitigate the impact of stem-loop primers on non-specific amplification during qPCR, only a portion of the reverse transcription product is typically added to the qPCR detection system, or the reverse transcription product is diluted prior to addition. While this approach enhances the specificity of miRNA detection, it also leads to considerable wastage of cDNA. The SPTM-PCR miRNA detection method, however, provides excellent specificity and effectively circumvents the non-specific amplification issues associated with stem-loop primers and upstream primers. Consequently, all cDNA can be utilized in the qPCR system, ensuring complete utilization of cDNA. The SE-SPTM-PCR miRNA detection platform not only combines sensitive and specific analytical performance but also simplifies and accelerates the analytical workflow, making it a cost-effective alternative. This represents a significant advancement over existing methods that rely on total RNA or total miRNA extraction and stem-loop RT-qPCR detection.

To evaluate the clinical applicability and effectiveness of the SE-SPTM-PCR miRNA detection platform, we applied this platform to analyze its diagnostic performance across three disease models. In the analysis of the diagnostic adaptability of SE-SPTM-PCR for CRC. Compared to two traditional miRNA detection methods, SE-SPTM-PCR exhibited higher sensitivity and specificity. This improvement may be attributed to its reduced susceptibility to background nucleic acid interference. It is important to note that the primary objective of this proof-of-concept clinical study was to evaluate the bioassay method rather than the biomarker itself. For this purpose, we collected four human miRNAs from the literature, which are strongly associated with CRC. Next, we employed SE-SPTM-PCR to evaluate the diagnostic performance of HCMV miRNAs in diagnosing HCMV reactivation in HSCT patients. SE-SPTM-PCR demonstrated a significant ability to distinguish between HSCT patients with HCMV reactivation and those without. Given the strong correlation between HCMV miRNA and DNA,

the detection of hcmv-miR-UL22A-5p in HSCT patients who are HCMV DNA-negative may be attributed to the higher sensitivity of miRNA compared to DNA. This finding suggests that future large-scale screening and validation of HCMV miRNA in the circulation of HSCT recipients using the developed SE-SPTM-PCR platform will contribute to the identification of effective HCMV miRNA candidate biomarkers for HCMV reactivation in HSCT recipients. We also evaluated the diagnostic performance of evb-miR-BART7-3p in NPC; SE-SPTM-PCR was able to significantly distinguish NPC patients from healthy individuals. In contrast, the diagnostic performance of evb-miR-BART7-3p in NPC was limited when utilizing traditional miRNA detection platforms. Previous studies have indicated that evb-miR-BART3-3p cannot differentiate NPC patients from healthy individuals[48]; however, the SE-SPTM-PCR platform successfully rediscovered the diagnostic value of evb-miR-BART3-3p in NPC.

These comparative results suggest that our SE-SPTM-PCR method has potential clinical adaptability and can enhance diagnostic performance in disease detection.

In summary, this study developed a SE-SPTM-PCR platform for the quantitative assessment of blood miRNAs, which offers unique advantages such as minimal background nucleic acid interference, high specificity, high sensitivity, a rapid analytical workflow, and low cost. A limitation of this method is its inability to directly address pre-analytical sample heterogeneity. The SE-SPTM-PCR significantly enhances specificity and sensitivity in clinical research, thereby improving the diagnostic performance of existing miRNA biomarkers for diseases. Given that SE-SPTM-PCR accurately reflects the clinical value of miRNA biomarkers, it can also measure a variety of other miRNAs beyond those examined in this study (Supplementary Table S10). This platform is poised to advance the field of liquid biopsy and has potential clinical applications that could enhance the quality of disease management. While this system is currently employed clinically only for patients with NPC, hematopoietic stem cell transplantation, and CRC, it is expected to be applicable across various domains of disease diagnosis and treatment, including early disease detection, provision of precise therapies, assessment of treatment responses, and early identification of disease recurrence.

## Methods

### Ethical statement
This study was approved by the Ethics Committee of Xiamen University (XDYX202507K24) and Fujian Medical University Union Hospital (2022KY167). All patients provided written informed consent.

### Materials
DNA and RNA oligonucleotides, biotin-modified and fluorescently labeled DNA, miRNA standards, peptide nucleic acids, and diethyl pyrocarbonate were obtained from Sangon Biotech Co., Ltd. Guanidine thiocyanate lysis reagent and proteinase K were sourced from Merck Bio. dNTPs and PCR reaction buffer were obtained from TaKaRa Bio. Trypsin solution was purchased from Shanghai BasalMedia Biotechnology Co., Ltd., and PBS buffer was procured from SIGMA.

### Procedure of SE-SPTM-PCR assay
Place 0.02 mg of hydrophilic streptavidin magnetic beads (NEB), for each extraction test, into an EP tube. After removing the bead storage solution using a magnetic stand, add biotin-labeled miRNA capture probes within a concentration range of 100 µM to 0.01 µM, ensuring a total volume of 100 µL for each extraction test. Subsequently, place the EP tube containing the beads and capture probes on an automatic rotary mixer and mix thoroughly for 10 min to facilitate the binding of biotin to streptavidin. Discard the supernatant, resuspend the labeled beads with Wash Buffer 1, discard the supernatant again, and repeat the washing process once more. Finally, resuspend the magnetic beads in the magnetic bead storage solution (0.5 M NaCl, 20 mM Tris-HCl

(pH = 7.5), 1 mM EDTA) at a volume of 10 μL per extraction test, and store at 4 °C for future use.

In columns 1 and 7 of the 96-well deep-well plate, add 100 μL of plasma sample, 24 μL of 5× lysis buffer, 10 μL of proteinase K, 6 μL of DEPC water, and 10 μL of magnetic beads labeled with miRNA capture probes. In columns 2, 3, 8, and 9 of the deep-well plate, add 100 μL of wash buffer 1 (0.5 M NaCl, 20 mM Tris-HCl, pH 7.5, 1 mM EDTA). Next, in columns 4 and 10 of the deep-well plate, add 100 μL of wash buffer 2 (0.15 M NaCl, 20 mM Tris-HCl, pH 7.5, 1 mM EDTA). Then, add 25 μL of elution buffer (10 mM Tris-HCl, pH 7.5, 1 mM EDTA) to columns 6 and 12 of the deep-well plate. Subsequently, place the deep-well plate on the automated extraction instrument (TIANLONG) and run the miRNA selective extraction according to the pre-set program. The extracted RNA samples are stored at -80 °C until use.

Add all selectively extracted and purified miRNAs to a mixture containing 2 μL of 10× reaction buffer, 0.4 μL of 10 μM reverse transcription primer, 0.05 μL of 10 mM dNTP, and 0.2 μL of reverse transcriptase (TransGen Biotech, 200 U/μL). Adjust the total volume of each reaction to 20 μL with nuclease-free water, and aliquot into 8-tube strips or 96-well plates. The reaction conditions are as follows: incubate in a thermal cycler at 16 °C for 30 min, followed by a reverse transcription reaction at 42 °C for 30 min, and then inactivate at 100 °C for 5 min. Store the reaction products at 4 °C. Subsequently, add all the products of the reverse transcription reaction to a mixture containing 0.5 μL of 10× reaction buffer, 1 μL of 10 μM upstream primer, 1 μL of 10 μM downstream primer, 1 μL of 10 μM TaqMan probe, 1 μL of 10 mM dNTP, and 0.2 μL of TaKaRa Taq™ Hot Start Version (TaKaRa, 5U/μL). Adjust the volume of each reaction to 20 μL using nuclease-free water, and dispense into 8-strip tubes or 96-well plates. Monitor the reaction using a real-time PCR detection system (BioRad) and incubated in a thermal cycler under the following conditions: enzyme activation at 95 °C for 5 min, followed by 45 cycles of denaturation at 95 °C for 10 s and annealing/extension at 55 °C for 30 s. A characteristic amplification curve was defined as a positive result. For each experiment, standard curves were generated from serially diluted synthetic target miRNAs of known concentration, both before and after the recovery step.

### The traditional miRNA detection process based on column extraction and RT-qPCR

Total RNA, including miRNA, was isolated from cell supernatants and plasma using the miRNeasy Serum/Plasma Kit (Qiagen) or the miRcute miRNA Isolation Kit (TIANGEN), according to the manufacturers' protocols. Following sample lysis, the exogenous spike-in control cel-miR-39-3p was added for quality control[57]. The extracted RNA was stored at −80 °C until further use.

The TaqMan® MicroRNA Reverse Transcription Kit, produced by Applied Biosystems, was employed to synthesize complementary DNA (cDNA). The process began by adding 5 μL of purified total RNA to a reaction mixture containing 1 μL of MultiScribe Reverse Transcriptase (50 U/μL), 1.5 μL of Reverse Transcription Buffer (10×), 0.19 μL of RNase Inhibitor (20 U/μL), 0.19 μL of dNTPs (100 mM, including dTTP), and 3 μL of 5×RT Primer (Thermofisher). The total volume was adjusted to 15 μL per reaction using nuclease-free water, and the mixture was distributed into 8-strip tubes or 96-well plates. The reaction conditions were as follows: incubation in a thermal cycler at 16 °C for 30 min, followed by a reverse transcription reaction at 42 °C for 30 min, and subsequent inactivation of the reaction at 85 °C for 5 min. The reaction products were stored at 4 °C.

Fluorescent quantitative PCR was conducted using the TaqMan® Universal Master Mix II, excluding the UNG synthesis kit from Applied Biosystems. A volume of 1.33 μL from the reverse transcription reaction product was combined with the TaqMan Small RNA Assay (20×) and mixed with 10 μL of the PCR Master Mix. Nuclease-free water was utilized to adjust the total volume of each reaction to 20 μL, which was

then aliquoted into 8-strip tubes or a 96-well plate. The reaction was monitored using a real-time PCR detection system (ABI 7500) and incubated in a thermal cycler under the following conditions: enzyme activation at 95 °C for 10 min, followed by 45 cycles of denaturation at 95 °C for 15 s and annealing/extension at 54 °C for 60 s. A characteristic amplification curve was defined as a positive result.

### Cell culture

Retrieve the cryovial containing MRC5 cells (ATCC, CCL-171) from the liquid nitrogen tank and quickly thaw it in a 37 °C water bath. Subsequently, centrifuge the cells at $1160 \times g$ for 3 min. Remove the supernatant and resuspend the cell pellet in 1 mL of complete medium, which consists of 90% MEM (Shanghai BasalMedia Biotechnology Co., Ltd.), 10% fetal bovine serum (Thermofisher), and 1% penicillin and streptomycin (Shanghai BasalMedia Biotechnology Co., Ltd.). Transfer the resuspended cells to a pre-prepared 10 cm cell culture dish (Thermofisher) containing 10 mL of complete medium, and incubate in a CO2 incubator at 37 °C with 5% CO2. Change the medium after 12 hours. Observe the cells under a standard inverted fluorescence microscope and passage them when they reach over 90% confluency.

### Viral infection and collection of cell supernant

When MRC5 cells reach the logarithmic growth phase, the HCMV virus (AD169) was added to the cell culture supernatant, and the culture plate was gently shaken to ensure thorough mixing. The cells were then incubated in a CO2 incubator, with their status monitored every other day. Infected cells could be observed under a microscope, displaying typical inclusion body structures. Once all cells were confirmed to be infected with HCMV, the culture was continued for an additional 24 h. Subsequently, the cell supernatant was collected, aliquoted into cryotubes, and stored at −80 °C for future use.

### Next-generation sequencing of non-specific amplification products of traditional stem-loop RT-qPCR

Nuclease-free water was added as a template to the BART12 stem-loop RT-qPCR assay to generate non-specific amplification products. DNA libraries were then constructed using the VAHTS Universal DNA Library Prep Kit for Illumina V2 (Vazyme), following the manufacturer's protocol. Library quality was assessed using the Agilent 2100 Bioanalyzer with the Bioanalyzer DNA Kit (Agilent) to assess fragment size distribution. Finally, libraries were sequenced on the NovaSeq 6000 platform (Illumina) with 150-base pair paired-end (PE150) reads.

### Next-generation sequencing of HCMV-positive simulated samples

A total of 30 plasma samples from healthy donors were divided into three age- and gender-matched groups to create a baseline matrix. Simulated specimens were then prepared by spiking plasma with the HCMV-containing culture supernatant and phosphate-buffered saline (PBS) at three different volume ratios: 90:10:0, 90:5:5, and 90:2.5:7.5 (plasma: HCMV supernatant: PBS). Total RNA, including the small RNA fraction, was isolated from 600 μL of each simulated specimen using the miRNeasy Serum/Plasma Advanced Kit (Qiagen) and eluted into 20 μL of nuclease-free water, following the manufacturer's instructions. RNA quality and concentration were assessed using the Agilent 2100 Bioanalyzer with the Small RNA Kit (Agilent Technologies). Subsequently, small RNA libraries were constructed from 5 μL of total RNA using the VAHTS Small RNA Library Prep Kit for Illumina V2 (Vazyme), according to the manufacturer's protocol. The amplified cDNA libraries were purified using VAHTSTM DNA Clean Beads (Vazyme). Library quality was validated on the Agilent 2100 Bioanalyzer to assess fragment size distribution and adapter dimer contamination. Finally, libraries were sequenced on the NovaSeq 6000 platform (Illumina) with 50-base pair single-end (SE50) reads.

## Clinical sample

In this research, plasma samples from colorectal cancer patients were collected for hsa-miR-92a-3p detection from the Xiamen Haicang Hospital. Plasma samples from individuals undergoing hematopoietic stem cell transplantation were obtained for hcmv-miR-UL22A-5p detection from the Fujian Medical University Union Hospital. Furthermore, plasma samples from patients diagnosed with nasopharyngeal carcinoma were obtained for EBV miRNAs detection from the Sun Yat-sen University Cancer Center. Plasma samples from healthy individuals were obtained from participants undergoing clinical and laboratory health check-ups.

## HCMV DNA detection

To extract HCMV DNA, use the magnetic bead extraction kit from Easunbio. Begin by removing the pre-packaged 96-deep well plate from the kit and centrifuge it at $2800 \times g$ for 1 min to ensure that the reagents settle at the bottom of the wells. Next, carefully remove the sealing film. Add 50 µL of elution buffer to the 6th and 12th wells of the deep well plate. For the 1st and 7th wells, add 20 µL of Proteinase K, 10 µL of magnetic beads, and 200 µL of plasma sample. In the 2nd and 8th wells, add 20 µL of Proteinase K and 200 µL of sample. Place the deep well plate onto the extraction instrument and perform the extraction according to the pre-set program. Upon completion of the run, immediately use the nucleic acid solution from the 6th and 12th wells. Finally, transfer any remaining nucleic acids to nuclease-free centrifuge tubes and store them at −80 °C.

Mix 2 µL of the extracted nucleic acid or standard with 40 µL of reaction buffer from the approved clinical-use Daan Gene Biotech kit and 3 µL of Taq enzyme. Distribute the total 45 µL sample volume of the qPCR reaction mix into 8-tube strips or 96-well plates, ensuring that the 8-tube strips are tightly sealed or the 96-well plates are adequately covered. Monitor the reaction using a real-time PCR detection system (BioRad) with the following thermal cycling conditions: enzyme activation at 93 °C for 3 min, followed by 40 cycles consisting of denaturation at 93 °C for 5 s and annealing/extension at 57 °C for 45 s.

## Statistics and reproducibility

Standard formulas were used to calculate the mean and standard deviation in Excel. The statistical results from multiple replicates in this study were presented as mean ± standard error. Differences between the two groups were compared using the two-tailed Mann-Whitney U test with Welch's correction ($P < 0.05$). ROC curve analyses were conducted using SPSS 27 to determine the AUC value. Chi-square tests and correlation analyses were performed using GraphPad Prism 10. No statistical method was used to predetermine sample size. No data were excluded from the analyses. The experiments were randomized. The analytical studies were not carried out blinded but included biological replicates to ensure minimal experimental bias.

## Data availability

The raw sequencing data of non-specific amplification products of traditional stem-loop RT-qPCR have been deposited into the Sequence Read Archive (SRA) database under accession number SRP679909. The raw sequencing data of HCMV-positive simulated samples have been deposited into Gene Expression Omnibus (GEO) database under accession number GSE321693. The data generated in this study are provided in the main text and Supplementary Information. Source data are provided as a Source Data file. Source data are provided with this paper.

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

## Acknowledgements

This work was supported by grants from the National Key Research and Development Program of China (2023YFD1800502 to S.Y.Z.), the National Natural Science Foundation of China (82070177 and U23A20419 to T.Y.), Major Science and Technology Project of Fujian Provincial Health Commission (2021ZD01006 to S.X.G., founded by Xiamen Municipal Health Commission), Natural Science Foundation of Xiamen, China (3502Z202373019 to S.Y.Z.), the Open Research Fund of State Key Laboratory of Vaccines for Infectious Diseases (2024SKLVDkf10 to Z.C.Z.), the Independent Research Project of State Key Laboratory of Vaccines for Infectious Diseases (2024SKLVDzy02 to T.D.L.), the Fundamental Research Funds for the Central Universities (20720220004 to N.S.X.), Project supported by the Natural Science Foundation of Fujian Province, China (2025J01747 to J.H.R., 2025J08343 to Z.C.Z.).

## Author contributions

S.Y.Z., T.Y., N.S.X., S.X.G., L.Q.T., and S.Z.H. conceived and supervised the project; Z.C.Z., W.D.H., J.H.R., L.Y., Q.R.Z., and Y.G. designed the research; Z.C.Z., W.D.H., J.H.R., L.Y., Q.R.Z., and Y.G. performed technology development, mechanistic study, analytical characterization, and clinical validation; J.W., J.Y.X., and L.S.L. conducted the methodology establishment and validation experiments; Y.M.C. constructed the colorectal cancer-related sample detection experiments; J.J.X.

performed the HSCT-related sample detection experiments; Z.Y.Z. conducted NPC-related sample detection experiments; T.D.L., J.Z. and J.D.H. assisted in the clinical studies; Z.C.Z. analysed the data and wrote the paper. All authors edited the paper.

## Competing interests

The authors declare no competing interests.
