## [Transparent Peer Review file · Nature Communications]

A selective enrichment and specific probe terminal mediated strategy for highly sensitive detection of microRNAs

Corresponding Author: Dr Shiyin Zhang

Version 0:

Reviewer comments:

Reviewer #1

(Remarks to the Author)

The manuscript by Zhong and colleagues reports on the Specific-Probe Terminal-Mediated RT qPCR quantification platform SE-SPTM-PCR, a novel detection platform integrating selective miRNA enrichment using locked nucleic acid probes with specific probe terminal mediated RT-qPCR.

One advantage that the authors claims is that compared to conventional stem-loop RT qPCR, SE-SPTM-PCR eliminates nonspecific amplification and achieves a 100-fold higher sensitivity.

A second one is that SE-SPTM-PCR enables high specificity and sensitivity in miRNA detection.

Several comments:

1 The authors compared this new method with the conventional stem-loop RT qPCR in HCMV negative and positive samples as well as colorectal cancer for one microRNA, the miR-92a-3p. The authors should use more microRNAs for this instance, at least 10 miRs and compare with both miRNA deep seq and conventional stem-loop RT qPCR.

2 Also, the authors should perform a study with results related to the base composition (G,A,C,T) and sequences characteristics of microRNAs.

3 The number of samples used for the results on diagnostic value to hcmv-miR-UL22A-5p for HCMV infection in hematopoietic stem cell transplantation is too low for drawing conclusions, 16 HCMV DNA positive and 11 DNA negative HSCT patients. The authors should expand these two cohorts largely.

4 As a general observation, the numbers of samples used and the number of microRNAs tested are generally small, I would increase both for each of the 3 main experiments: the sensitivity/specificity experiment, the diagnostic performances in colorectal cancer and the HCMV infection in hematopoietic stem cell transplantation.

Reviewer #2

(Remarks to the Author)

This manuscript describes the development and validation of SE-SPTM-PCR, a novel, highly sensitive platform for detecting circulating microRNAs (miRNAs). The technical innovation, which integrates selective enrichment with a unique probe-mediated RT-qPCR strategy, is sound, well-reasoned, and of high technical interest to the field. The method successfully addresses critical analytical challenges in conventional miRNA quantification, such as non-specific amplification and matrix interference. However, the study is primarily a method validation paper and, as the authors note, it focuses on enhancing the performance of existing biomarkers. The current findings do not yield new biological insights into disease mechanisms or de novo biomarker discovery. Specific concerns follow:

1. The selection of hsa-miR-92a-3p for colorectal cancer (CRC) validation is insufficiently justified in the context of validating a platform. The stated rationale is its "well-established association with CRC". This gives the appearance that a convenient,

favorable miRNA was chosen to maximize the performance gap between the new and conventional methods. To definitively support the claim that SE-SPTM-PCR is a broadly transformative platform, the authors must address the generalizability of their findings. It is doubtful how much the superior diagnostic performance applies to other miRNAs. A comprehensive profiling experiment is strongly recommended using both conventional RT-qPCR and SE-SPTM-PCR. This is crucial for demonstrating the platform's utility beyond a hand-picked target.

2. A critical omission is the lack of a clear description regarding the internal control used for quantifying circulating miRNAs in the clinical validation studies. Circulating miRNA assays require normalization against a stable reference to correct for pre-analytical variations and variations in extraction/RT efficiency. The authors must explicitly state which internal control/housekeeping RNA was used (or why one was not used, though this is highly discouraged) and provide supporting data on its stability and recovery within the SE-SPTM-PCR system. Without proper normalization, the quantitative comparisons and diagnostic conclusions are methodologically compromised.

3. The authors use HCMV DNA detection as the "gold standard" to define positive/negative cases. If the established, approved HCMV-DNA assay already serves as the high-performance diagnostic tool for monitoring, the clinical need for an HCMV miRNA assay is questionable. The authors need to provide a more straightforward, more compelling clinical argument for how miRNA detection, even with enhanced sensitivity, will replace or add superior value to the existing, established DNA gold standard for patient management.

4. A major practical obstacle in the clinical translation of circulating miRNA assays is the strict post-phlebotomy handling (e.g., time to centrifugation) required to prevent blood cell lysis and the subsequent artificial release of intracellular miRNAs into the plasma. This pre-analytical variability is a significant barrier to practical use. The developed SE-SPTM-PCR system is an analytical improvement but does not inherently address this pre-analytical issue. The Discussion section must acknowledge this crucial limitation. Ideally, the authors should present pilot data to demonstrate whether their selective enrichment step can mitigate the impact of delayed sample processing on their target miRNA levels.

5. The ROC curve shown in Figure 6d for the HCMV analysis is excessively smooth, especially given the small sample size. A proper ROC curve plotted from discrete clinical data points should be a stepped line.

6. Figure 6b shows that conventional RT-qPCR resulted in all samples (16/16 DNA-positive and 11/11 DNA-negative) being scored as "hcmv-miR-UL22A-5p pos." If all samples were positive, how was the distinction between "positive" and "negative" made (i.e., what was the Ct or Cq cutoff value)? The authors must clearly define and justify the cutoff used to binarize the conventional RT-qPCR results for this specific miRNA.

7. Please shorten the Introduction by focusing more concisely on the high-level challenges in circulating miRNA detection, thus improving the paper's flow and readability.

Version 1:

Reviewer comments:

Reviewer #1

(Remarks to the Author)

The authors performed a revision according to the reviewers comments.

This is of interest for the large spectrum of readers of the journal.

Reviewer #2

(Remarks to the Author)

The authors properly answered my comments.

Reviewer #1 (Remarks to the Author):

The manuscript by Zhong and colleagues reports on the Specific-Probe Terminal-Mediated RT qPCR quantification platform SE-SPTM-PCR, a novel detection platform integrating selective miRNA enrichment using locked nucleic acid probes with specific probe terminal mediated RT-qPCR.

One advantage that the authors claims is that compared to conventional stem-loop RT qPCR, SE-SPTM-PCR eliminates nonspecific amplification and achieves a 100-fold higher sensitivity.

A second one is that SE-SPTM-PCR enables high specificity and sensitivity in miRNA detection.

Response: We appreciate your accurate summary of our study's research value and technical innovations, particularly your recognition of the SE-SPTM-PCR platform's novelty. We concur with your assessment of these advantages and have now supplemented key experimental evidence to reinforce our claims.

Several comments:

1 The authors compared this new method with the conventional stem-loop RT qPCR in HCMV negative and positive samples as well as colorectal cancer for one microRNA, the miR-92a-3p. The authors should use more microRNAs for this instance, at least 10 miRs and compare with both miRNA deep seq and conventional stem-loop RT qPCR.

Response: Thank you for this excellent and essential suggestion. We completely agree that rigorously evaluating our platform's generalizability requires comprehensive benchmarking against conventional stem-loop RT-qPCR and miRNA deep sequencing across a broader miRNA panel. To this end, we have conducted extensive additional experiments.

Our comparative analysis now encompasses 12 miRNAs across three distinct disease models: four human miRNAs implicated in colorectal cancer (CRC) (hsa-miR-92a-3p, hsa-miR-320a, hsa-miR-19a-3p, hsa-miR-423-5p); five viral miRNAs associated with HCMV reactivation post-hematopoietic stem-cell transplantation (HSCT) (hcmv-miR-US25-1-5p, hcmv-miR-UL22A-5p, hcmv-miR-US5-2-3p, hcmv-miR-US4-3p, hcmv-miR-US25-2-3p); and three EBV miRNAs relevant to nasopharyngeal carcinoma (NPC) (ebv-miR-BART7-3p, ebv-miR-BART2-5p, ebv-miR-BART3-3p).

The CRC-associated miRNAs were selected based on established diagnostic literature¹⁻⁴ and prioritized for their relatively high abundance in serum. For HSCT patient HCMV reactivation-associated HCMV miRNAs, we included hcmv-miR-UL22A-5p (previously reported as non-diagnostic for HCMV reactivation of post-HSCT)⁵ and four additional other HCMV-encoded miRNAs⁶. For NPC-associated EBV miRNAs, the panel comprised both established diagnostic markers for NPC (ebv-miR-BART7-3p, -BART2-5p)^{7, 8} and a reported non-diagnostic marker (ebv-miR-BART3-3p)⁷.

In CRC diagnosis, both conventional stem-loop RT-qPCR and SE-SPTM-PCR methods

significantly distinguished patients from healthy controls. However, SE-SPTM-PCR demonstrated overall superior diagnostic performance across the four miRNAs (Fig. 1a-l in response letter). For hsa-miR-92a-3p and hsa-miR-320a, SE-SPTM-PCR provided more significant discrimination and higher AUC values compared to conventional methods (Fig. 1a-f in response letter; Fig. 5b-d in manuscript; supplementary Fig. S24a-c). For hsa-miR-19a-3p and hsa-miR-423-5p, SE-SPTM-PCR outperformed one conventional protocol (using QIAGEN column extraction and stem-loop primer RT-qPCR) and performed equivalently to another (using TIANGEN column extraction and stem-loop primer RT-qPCR) (Fig. 1g-l in response letter; Supplementary Fig. S24d-i).

Superior diagnostic performance for CRC depends on two key factors: a reliable detection method and diagnostic targets with inherent disease-discriminating capability. To specifically evaluate the reliability of the detection method, we designed the following experiments. The four miRNAs analyzed are endogenous to humans and exhibit baseline expression in healthy individuals. Therefore, their diagnostic utility for CRC primarily depends on differential expression abundance. Accordingly, undiluted healthy human serum was used to simulate patient samples, while a 4-fold PBS dilution of the same serum was used to simulate healthy control samples. Ten technical replicates were prepared for each condition. This experimental design aimed to evaluate and compare the ability of the SE-SPTM-PCR method versus conventional stem-loop primer RT-qPCR to distinguish these target abundance differences. The results demonstrated that SE-SPTM-PCR was significantly superior to conventional methods in distinguishing the simulated patient samples from the healthy controls (Fig. 2a, b in response letter). For all four miRNAs analyzed, SE-SPTM-PCR achieved perfect AUC scores of 1.0 (Fig. 2a in response letter). In contrast, the conventional method yielded AUC values ranging from only 0.53 to 0.63 (Fig. 2b in response letter). These findings indicate the inherent superiority of the SE-SPTM-PCR technique. The variation in diagnostic improvement across miRNAs (Fig. 1 in response letter) may originate from differences in target miRNA abundance between actual patients and healthy controls,

combined with differential susceptibility of these miRNAs to background nucleic acid interference. This study focuses on assessing the analytical performance of this platform using validated targets. In summary, results from the simulated sample experiments demonstrate the robust capability of SE-SPTM-PCR to discriminate target abundance differences.

Fig. 1 Diagnostic performance comparison of SE-SPTM-PCR versus conventional miRNA detection methods for CRC diagnosis using hsa-miR-92a-3p (a-c), hsa-miR-320a (d-f), hsa-miR-19a-3p (g-i) and hsa-miR-423-5p (j-l). a, d, g, j Diagnostic performance of conventional scheme 1 (TIANGEN column extraction combined with stem-loop primer RT-qPCR). b, e, h, k Diagnostic performance of conventional scheme 2 (QIAGEN column extraction combined with stem-loop primer RT-qPCR). c, f, i, l Diagnostic accuracy of the SE-SPTM-PCR based detection system. CRC patients, n=48; healthy controls, n=48. **P < 0.01, ***P < 0.001, ****P < 0.0001, “ns” means not significant.

Fig. 2 Diagnostic performance comparison of SE-SPTM-PCR and conventional miRNA detection method using simulated patients and simulated healthy controls. a The SE-SPTM-PCR method was used to detect human miRNAs and the results were analyzed by ROC curve. **b** The conventional miRNA detection method was used to detect human miRNAs and the results were analyzed by ROC curve. CRC, colorectal cancer. HC, healthy control.

For HCMV reactivation monitoring, SE-SPTM-PCR clearly differentiated HCMV-DNA-positive from HCMV-DNA-negative transplant patients across all five

HCMV miRNAs, with AUCs >0.80 (Fig. 3b, c, e, f, h, i, k, l, n, o in response letter; Fig. 6c, d in manuscript; supplementary Fig. S25 b, c, e, f, h, i, k, l). In contrast, conventional stem-loop RT-qPCR failed to distinguish HCMV DNA-positive from DNA-negative patients for three of the five miRNAs (Fig. 3a, g, j in response letter; Fig. 6b in manuscript; supplementary Fig. S25 d, g). For the remaining two miRNAs, the discrimination achieved by the conventional method was less significant than with SE-SPTM-PCR (Fig. 3d, m in response letter; supplementary Fig. S25 a, j).

Fig. 3 Diagnostic performance comparison of SE-SPTM-PCR and conventional miRNA detection system for monitoring HCMV reactivation in HSCT recipients. a, d, g, j, m Detection in HCMV DNA positive and negative HSCT patients using conventional RT-qPCR. b, e, h, k, n Detection in HCMV DNA positive and negative HSCT patients using SE-SPTM-PCR. c, f, i, l, o ROC curve analysis of hcmv-miR-US25-1-5p, hcmv-miR-UL22A-5p, hcmv-miR-US25-2-3p, hcmv-miR-US4-3p, hcmv-miR-US5-2-3p. The orange line represents SE-SPTM-PCR, and the

green line represents conventional RT-qPCR.

In NPC diagnosis, conventional stem-loop RT-qPCR, while showing statistically significant differences for the three EBV miRNAs, provided limited clinical discrimination between patients and healthy controls (Fig. 4a, c, e, g in response letter; Fig. 7b, d, f, h in manuscript). In contrast, SE-SPTM-PCR not only yielded statistical significance but also provided clear, clinically relevant discrimination (Fig. 4b, d, f, h in response letter; Fig. 7c, e, g, i in manuscript).

Fig. 4 Diagnostic performance evaluation of SE-SPTM-PCR versus conventional RT-qPCR for NPC detection using EBV miRNAs. a-f Clinical validation results: a Detection of ebv-miR-

BART7-3p by conventional RT-qPCR. b Detection of ebv-miR-BART7-3p by SE-SPTM-PCR. c Detection of ebv-miR-BART3-3p by conventional RT-qPCR. d Detection of ebv-miR-BART3-3p by SE-SPTM-PCR. e Detection of ebv-miR-BART2-5p by conventional RT-qPCR. f Detection of ebv-miR-BART2-5p by SE-SPTM-PCR. g, h Diagnostic performance of the EBV miRNA system based on conventional RT-qPCR and SE-SPTM-PCR for NPC (NPC, n=40; healthy individuals, n=40).

In the comparison of miRNA deep seq and SE-SPTM-PCR. Given the substantial sample requirements of deep sequencing and the limited availability of clinical specimens, we designed a controlled model to evaluate sensitivity to subtle expression changes. Healthy human plasma was spiked with exogenous HCMV culture supernatant and subjected to a 2-fold gradient dilution to simulate minor differences in viral load. Analysis of the same five HCMV miRNAs revealed that while both deep sequencing and SE-SPTM-PCR detected the dilution trend (Fig. 5 in response letter; supplementary Fig. S26), SE-SPTM-PCR was significantly superior in resolving 2-fold and 4-fold differences, as reflected by higher AUC values in ROC analysis (Fig. 5a, b in response letter; supplementary Fig. S26a, b).

Fig. 5 Diagnostic performance comparison of SE-SPTM-PCR and miRNA deep sequencing using 2-fold gradient dilutions of simulated HCMV viral infection samples. a The SE-SPTM-PCR method was used to detect HCMV miRNAs and the results were analyzed by ROC curve. **b** The deep sequencing method was used to detect HCMV miRNAs and the results were analyzed by ROC curve.

In summary, SE-SPTM-PCR demonstrates superior performance to both conventional stem-loop RT-qPCR and miRNA deep sequencing for diagnosing diseases associated with human and viral miRNAs.

2 Also, the authors should perform a study with results related to the base composition (G, A, C, T) and sequences characteristics of microRNAs.

Response: We thank you for this insightful and valuable suggestion. We agree that analyzing the base composition and sequences characteristics of microRNAs is crucial for comprehensively evaluating the general applicability and potential limitations of our SE-SPTM-PCR platform. In direct response to your comment, we have conducted a systematic bioinformatic analysis of the sequence and thermodynamic properties of all 12 miRNAs included in our study.

We analyzed the following characteristics for all 12 miRNAs: sequence length, GC content, predicted melting temperature (T_m), free energy (ΔG , kcal/mol), and—where applicable—loop T_m and loop free energy (Loop ΔG , kcal/mol). The miRNA lengths ranged from 20 to 24 nucleotides (nt). The GC content ranged from 34.78% to 66.67%. The predicted T_m values spanned from 59.2 to 80.3. The ΔG values ranged from -35.6 to -51.5 kcal/mol. Half of the miRNAs (6/12) were predicted to lack a canonical loop structure. For the remaining half that possessed a loop, the predicted loop T_m values ranged from 7 to 77.6. The loop ΔG values for these miRNAs ranged from -2.6 to 2.6 kcal/mol (Table 1 in response letter; supplementary Table S10). These results demonstrate that SE-SPTM-PCR is suitable for miRNAs spanning a broad range of lengths and GC contents, indicating its potential for future extension to a wider array of miRNA targets (Line 595 in manuscript).

Table 1 Analysis of miRNA characteristics included in this study

miRNA	Sequence	Length (nt)	GC%	Tm	ΔG (Kcal/mol)	Loop Tm	Loop ΔG (Kcal/mol)	
Human	hsa-miR-92a-3p	UAUUGCACUUGUCCCG GCCUGU	22	54.55%	73.6	-44.8	/	1.2
	hsa-miR-320a	AAAAGCUGGGUUGAGA GGGCGA	22	54.55%	74.4	-46.1	/	1.8
	hsa-miR-19a-3p	UGUGCAAUUCUAUGCA AAACUGA	23	34.78%	66.2	-40.1	77	-2.6
	hsa-miR-423-5p	UGAGGGGCAGAGA GCGAGACUUU	24	54.17%	75.1	-47.5	7	0.9
HCMV	hcmv-miR-US25-1-5p	AACCGCUCAGUGGCUC GGACC	21	66.67%	75.4	-45.0	67	-2.4

	hcmv-miR-UL22A-5p	UAACUAGCCUCCCCGU GAGA	20	50.00%	62.3	-37.5	/	2.4
	hcmv-miR-US5-2-3p	UAUGAUAGGUGUGACG AUGUCU	22	40.91%	59.2	-35.6	/	2.3
	hcmv-miR-US4-3p	UGACAGCCCCGCUACAC CUCU	20	60.00%	67.6	-39.3	/	2.6
	hcmv-miR-US25-2-3p	AUCCACUUGGAGAGCU CCCGCGGU	24	62.5%	80.3	-51.5	27	-0.2
	ebv-miR-BART3-3p	CGCACCACUAGUCACC AGGUGU	22	59.09%	69.4	-40.7	56	-1.8
EBV	ebv-miR-BART7-3p	CAUCAUAGUCCAGUGU CCAGGG	22	54.55%	66.4	-39.4	/	1.0
	ebv-miR-BART2-5p	UAUUUUCUGCAUUCGC CCUUGC	22	45.45%	71.6	-44.7	49	-0.9

nt: nucleotides. “Tm” means melting temperature. ΔG : Gibbs free-energy change. “/” means not applicable. “HCMV” means Human Cytomegalovirus. “EBV” means Epstein–Barr virus. Tm, ΔG , Loop Tm and Loop ΔG are calculated using Oligo 6 software.

3 The number of samples used for the results on diagnostic value to hcmv-miR-UL22A-5p for HCMV infection in hematopoietic stem cell transplantation is too low for drawing conclusions, 16 HCMV DNA positive and 11 DNA negative HSCT patients. The authors should expand these two cohorts largely.

Response: We sincerely thank you for raising this important point regarding the sample size for evaluating hcmv-miR-UL22A-5p in HCMV infection among HSCT patients. We fully agree that a larger cohort is essential for drawing robust conclusions regarding its diagnostic value.

In direct response to this suggestion, we have made substantial efforts to expand our patient cohorts. We acknowledge the inherent challenge in rapidly acquiring samples from this specific population, as HSCT patients are less prevalent than those with many other diseases (such as influenza). Through close collaboration with our partner hospital, we have exhaustively gathered all available, qualifying samples within the study period.

Consequently, we have successfully expanded both cohorts. The HCMV DNA-positive HSCT cohort was increased from 16 to 32 patients, and the DNA-negative cohort was increased from 11 to 32 patients. This expansion doubles the positive cohort and nearly triples the negative cohort, thereby significantly enhancing the statistical power of our analysis.

All diagnostic performance metrics for hcmv-miR-UL22A-5p have been re-calculated using this expanded dataset of 64 patients and are updated in the revised manuscript (Fig. 3d-f in response letter; supplementary Fig. S25a-c). Although conventional RT-qPCR significantly distinguished HCMV DNA-negative from DNA-positive HSCT patients (Fig. 3d in response letter; supplementary Fig. S25a), the SE-SPTM-PCR method demonstrated superior discriminatory power (Fig. 3e in response letter; supplementary Fig. S25b), with an AUC of 0.95 in ROC curve analysis (Fig. 3f in response letter; supplementary Fig. S25c).

4 As a general observation, the numbers of samples used and the number of microRNAs tested are generally small, I would increase both for each of the 3 main experiments: the sensitivity/specificity experiment, the diagnostic performances in colorectal cancer and the HCMV infection in hematopoietic stem cell transplantation.

Response: We sincerely thank you for this overarching and constructive suggestion regarding the scale of our study. We fully agree that increasing both the number of samples and the number of microRNAs tested is crucial for strengthening the statistical power and generalizability of the conclusions drawn from our three main experimental applications.

In direct response to your recommendation, we have made substantial efforts to expand the scope of each part of our study. The specific enhancements are detailed below:

1. Diagnostic application in CRC:

miRNAs: The panel of human miRNAs analyzed was expanded from 1 to 4.

Samples: The patient cohort was increased from 33 to 48 CRC patients, with a matched healthy control cohort also expanded from 36 to 48 individuals.

2. Diagnostic application for HCMV reactivation in HSCT:

miRNAs: The panel of HCMV-derived miRNAs analyzed was expanded from 1 to 5.

Samples: The cohort of HCMV DNA-positive HSCT patients was increased from 16 to 32, and the HCMV DNA-negative HSCT patient cohort was increased from 11 to 32.

3. Diagnostic application in NPC:

miRNAs: The panel of 3 EBV-encoded miRNAs was maintained.

Samples: The NPC patient cohort was increased from 27 to 40 patients, and the matched healthy control cohort was substantially expanded from 8 to 40 individuals.

Utilizing these expanded datasets, we have systematically re-performed all relevant

analyses. The new results support the original conclusion, confirming that the SE-SPTM-PCR method offers superior diagnostic performance over conventional method for reported miRNAs. For CRC diagnosis, the SE-SPTM-PCR method detected 4 miRNAs (hsa-miR-92a-3p, hsa-miR-320a, hsa-miR-19a-3p, and hsa-miR-423-5p), yielding AUC values of 0.85, 0.70, 0.65, and 0.68, respectively. These AUCs were higher than those obtained with the conventional method (QIAGEN column extraction followed by stem-loop primer RT-qPCR), which were 0.80, 0.67, 0.59, and 0.64 (Fig. 1 in response letter; Fig. 5 in manuscript; supplementary Fig. S24). For diagnosing HCMV reactivation in HSCT patients, SE-SPTM-PCR analysis of 5 miRNAs (hcmv-miR-US25-1-5p, hcmv-miR-UL22A-5p, hcmv-miR-US5-2-3p, hcmv-miR-US4-3p, and hcmv-miR-US25-2-3p) produced AUC values of 0.96, 0.95, 0.97, 0.82, and 0.90. All these values exceeded the corresponding AUCs from conventional method (0.81, 0.89, 0.65, 0.59, and 0.68) (Fig. 3 in response letter; Fig. 6 in manuscript; supplementary Fig. S25). In NPC diagnosis, SE-SPTM-PCR detection of 3 miRNAs (ebv-miR-BART7-3p, ebv-miR-BART2-5p, and ebv-miR-BART3-3p) achieved perfect AUCs of 1.00, surpassing the values from conventional method (0.85, 0.77, and 0.96) (Fig. 4 in response letter; Fig. 7 in manuscript).

Reviewer #2 (Remarks to the Author):

This manuscript describes the development and validation of SE-SPTM-PCR, a novel, highly sensitive platform for detecting circulating microRNAs (miRNAs). The technical innovation, which integrates selective enrichment with a unique probe-mediated RT-qPCR strategy, is sound, well-reasoned, and of high technical interest to the field. The method successfully addresses critical analytical challenges in conventional miRNA quantification, such as non-specific amplification and matrix interference. However, the study is primarily a method validation paper and, as the authors note, it focuses on enhancing the performance of existing biomarkers. The current findings do not yield new biological insights into disease mechanisms or de novo biomarker discovery. Specific concerns follow:

Response: We sincerely thank you for your high evaluation of the SE-SPTM-PCR technology platform, especially the recognition of the rationality of its technological innovation, the rigor of the demonstration and the technical value of the field. We also thank you for clearly pointing out the success of this study on key analytical challenges such as non-specific amplification and matrix interference. These are certainly great encouragement to our team.

At the same time, we also appreciate your valuable comments on the positioning and biological contribution of our research, and we fully understand and agree with your views. The main goal of this study is to complete a methodological verification, aiming to systematically solve the core bottlenecks in existing miRNA detection technologies. We chose known miRNA biomarkers with clear clinical associations as validation objects to demonstrate the fundamental improvement in analytical performance (such as sensitivity, specificity, and robustness) of the SE-SPTM-PCR platform. While this study did not reveal novel disease mechanisms or identify new biomarkers, it demonstrated that the SE-SPTM-PCR detection method can restore diagnostic significance to miRNAs—including hcmv-miR-UL22A-5p and ebv-miR-BART3-3p—

previously regarded as lacking diagnostic value. We believe that the establishment of a solid and reliable technical foundation is the primary prerequisite for any subsequent biological discovery and clinical transformation. Your comments also point out the important expansion direction of the future influence of this work.

1. The selection of hsa-miR-92a-3p for colorectal cancer (CRC) validation is insufficiently justified in the context of validating a platform. The stated rationale is its "well-established association with CRC". This gives the appearance that a convenient, favorable miRNA was chosen to maximize the performance gap between the new and conventional methods. To definitively support the claim that SE-SPTM-PCR is a broadly transformative platform, the authors must address the generalizability of their findings. It is doubtful how much the superior diagnostic performance applies to other miRNAs. A comprehensive profiling experiment is strongly recommended using both conventional RT-qPCR and SE-SPTM-PCR. This is crucial for demonstrating the platform's utility beyond a hand-picked target.

Response: Thank you for this critical and insightful comment regarding the selection of hsa-miR-92a-3p and the need to demonstrate the generalizability of the SE-SPTM-PCR platform. We fully agree that relying on a single, well-established miRNA is insufficient to support the claim of a broadly applicable platform, as it could introduce selection bias.

In direct response to your suggestion and that of Reviewer #1, we have undertaken a comprehensive effort to address the issue of generalizability. To move beyond a single target, we conducted a literature review to identify human miRNAs with reported diagnostic potential for CRC¹⁻⁴. From this list, we selected four miRNAs—hsa-miR-92a-3p, hsa-miR-320a, hsa-miR-19a-3p, and hsa-miR-423-5p—based primarily on their relatively high abundance in serum. No further filtering based on sequence or thermodynamic properties that might favor our platform was applied. A head-to-head comparative analysis of these four miRNAs was performed using both conventional stem-loop RT-qPCR and our SE-SPTM-PCR platform. This analysis was conducted on an expanded cohort comprising 48 CRC patients and 48 age- and sex-matched healthy controls. The results demonstrated that both conventional methods and the SE-SPTM-PCR platform could significantly distinguish CRC patients from healthy controls across all tested miRNAs (Fig. 1 in response letter; Fig. 5 in manuscript; supplementary Fig.

S24). Overall, the SE-SPTM-PCR platform demonstrated superior diagnostic performance. This superiority was most pronounced for hsa-miR-92a-3p and hsa-miR-320a, for which SE-SPTM-PCR achieved higher AUC values and provided more significant separation between groups (Fig. 1a-f in response letter; Fig. 5b-d in manuscript; supplementary Fig. S24a-c). For hsa-miR-19a-3p and hsa-miR-423-5p, the performance of SE-SPTM-PCR was superior to one conventional method (QIAGEN column extraction + stem-loop RT-qPCR) and comparable to the other (TIANGEN column extraction + stem-loop RT-qPCR) (Fig. 1g-l in response letter; Supplementary Fig. S24d-i).

Superior diagnostic performance for CRC depends on two key factors: a reliable detection method and diagnostic targets with inherent disease-discriminating capability. To specifically evaluate the reliability of the detection method, we designed the following experiments. The four miRNAs analyzed are endogenous to humans and exhibit baseline expression in healthy individuals. Therefore, their diagnostic utility for CRC primarily depends on differential expression abundance. Accordingly, undiluted healthy human serum was used to simulate patient samples, while a 4-fold PBS dilution of the same serum was used to simulate healthy control samples. Ten technical replicates were prepared for each condition. This experimental design aimed to evaluate and compare the ability of the SE-SPTM-PCR method versus conventional stem-loop primer RT-qPCR to distinguish these target abundance differences. The results demonstrated that SE-SPTM-PCR was significantly superior to conventional methods in distinguishing the simulated patient samples from the healthy controls (Fig. 2a, b in response letter). For all four miRNAs analyzed, SE-SPTM-PCR achieved perfect AUC scores of 1.0 (Fig. 2a in response letter). In contrast, the conventional method yielded AUC values ranging from only 0.53 to 0.63 (Fig. 2b in response letter). These findings indicate the inherent superiority of the SE-SPTM-PCR technique. The variation in diagnostic improvement across miRNAs (Fig. 1 in response letter) may originate from differences in target miRNA abundance between actual patients and healthy controls, combined with differential susceptibility of these miRNAs to background nucleic acid

interference. This study focuses on assessing the analytical performance of this platform using validated targets. In summary, results from the simulated sample experiments demonstrate the robust capability of SE-SPTM-PCR to discriminate target abundance differences.

Importantly, the advantage of SE-SPTM-PCR is not limited to CRC or to human miRNAs. Parallel validations in other disease models reinforce this general utility. For diagnosing HCMV reactivation in HSCT patients, SE-SPTM-PCR outperformed conventional method for HCMV miRNAs (Fig. 3 in response letter; Fig. 6 in manuscript; supplementary Fig. S25). For NPC diagnosis, SE-SPTM-PCR showed superior performance for EBV miRNAs (Fig. 4 in response letter; Fig. 7 in manuscript).

Furthermore, to substantiate the platform's broad applicability, we conducted a bioinformatic analysis of the sequence and thermodynamic characteristics—including length, GC content, and T_m —of all 12 miRNAs used in this study. These miRNAs span a wide range of these parameters (Table 1 in response letter; supplementary Table S10), suggesting that the SE-SPTM-PCR method is suitable for detecting miRNAs with diverse properties and is not limited to a narrowly selected subset.

In conclusion, our data robustly demonstrate that the performance advantage of SE-SPTM-PCR extends across: (1) multiple miRNAs selected without platform-specific bias, (2) different disease contexts (CRC, HCMV reactivation, and NPC), and (3) miRNAs with diverse sequence properties.

2. A critical omission is the lack of a clear description regarding the internal control used for quantifying circulating miRNAs in the clinical validation studies. Circulating miRNA assays require normalization against a stable reference to correct for pre-analytical variations and variations in extraction/RT efficiency. The authors must explicitly state which internal control/housekeeping RNA was used (or why one was not used, though this is highly discouraged) and provide supporting data on its stability and recovery within the SE-SPTM-PCR system. Without proper normalization, the quantitative comparisons and diagnostic conclusions are methodologically compromised.

Response: Thank you for raising this critical methodological point regarding the use of an internal control for circulating miRNA quantification. We sincerely apologize for the lack of clarity on this important issue in our initial manuscript. We fully agree that proper normalization is essential for robust quantitative comparisons. Below, we provide a detailed description of our control strategy.

Regarding conventional miRNA extraction and detection methods, we acknowledge that the use of endogenous miRNAs as reference genes remains controversial. For instance, while some studies suggest hsa-miR-16-5p is suitable as an endogenous reference for disease diagnosis^{9, 10}, others report its dysregulation in conditions such as breast cancer and chordoma^{11, 12}, thereby undermining its reliability as a stable control. In contrast, exogenous spike-in controls, such as cel-miR-39-3p, offer a controllable concentration and are not influenced by intrinsic biological sample factors, providing a more stable baseline for normalization¹³. To monitor and correct for variations in miRNA extraction and reverse transcription efficiency, the exogenous spike-in control cel-miR-39-3p was added to the sample lysate immediately after lysis in our standard column-based extraction protocol (Lines 656-657 in manuscript). This step was performed under strictly controlled and consistent sample collection and pre-processing conditions. Relevant data supporting this normalization approach are now presented in the revised manuscript (Fig. 5b, c in manuscript; Supplementary Fig. S24a, b, d, e, g,

h).

For the SE-SPTM-PCR platform, we fully agree with your perspective on using reference to calibrate the target miRNA detection. In the initial phase of this study, we similarly explored a calibration strategy using an exogenous internal reference (cel-miR-39-3p). Specifically, we prepared magnetic beads designed to co-capture the endogenous target (hsa-miR-92a-3p) and the exogenous internal reference (cel-miR-39-3p). Following plasma lysis, a known quantity of the cel-miR-39-3p standard was spiked in to enable the co-extraction of both miRNAs. This approach allowed us to assess the feasibility of using the cel-miR-39-3p signal to correct for inter-sample variations. Plasma was collected from eight healthy donors. For each donor plasma sample, seven technical replicates were prepared. All replicates were analyzed using SE-SPTM-PCR. The results showed that the coefficient of variation (CV) for hsa-miR-92a-3p was low (<1.5%) both with and without normalization (Fig. 6a, b in response letter). However, the normalized CV was slightly higher than the non-normalized CV (Fig. 6a, b in response letter). This may be because the co-extraction of the abundant endogenous hsa-miR-92a-3p compromised the consistent recovery of the spiked-in cel-miR-39-3p standard. Given the potential for interference during co-extraction and the observation that independent target miRNA extraction yielded both excellent stability (CV <1.5%, Fig. 6a in response letter) and higher recovery in complex samples than traditional methods (Fig. 6c-e in response letter, supplementary Fig. S13a-c), we adopted a single-plex extraction and detection strategy for subsequent studies. Of course, as you rightly pointed out, monitoring variability throughout the entire miRNA detection process is crucial. As an alternative quality control measure, standard curves were generated for each experiment using serial dilutions of a synthetic target miRNA of known concentration. This was performed both pre- and post-nucleic acid recovery (Lines 649-651 in manuscript). This method allows for process efficiency monitoring and ensures the accuracy and reliability of quantification.

Fig. 6 The stability, anti-interference ability and recovery rate of the self-built method were evaluated. **a** CV of endogenous hsa-miR-92a-3p detected by SE-SPTM-PCR method in different plasma samples. **b** CV of hsa-miR-92a-3p corrected by cel-miR-39-3p. **c** Plasma samples under hemolytic, normal, and lipemic conditions. **d** Endogenous miRNA levels in plasma samples extracted by the three methods. **e** Recovery rates of exogenous miRNA in plasma samples extracted by the three methods. CV, coefficient of variation.

3. The authors use HCMV DNA detection as the "gold standard" to define positive/negative cases. If the established, approved HCMV-DNA assay already serves as the high-performance diagnostic tool for monitoring, the clinical need for an HCMV miRNA assay is questionable. The authors need to provide a more straightforward, more compelling clinical argument for how miRNA detection, even with enhanced sensitivity, will replace or add superior value to the existing, established DNA gold standard for patient management.

Response: Thank you for this insightful and critical question regarding the clinical necessity of an HCMV miRNA assay given the established role of DNA detection. We agree that demonstrating superior clinical value is essential.

Monitoring for HCMV reactivation is critical in the clinical management of HSCT patients, as it is associated with severe complications including HCMV disease, graft-versus-host disease (GVHD), increased opportunistic infections, and bone marrow suppression¹⁴⁻¹⁶. Historically, HCMV reactivation was monitored using immunofluorescence to detect the pp65 antigen in blood cells. This was subsequently superseded by more sensitive HCMV DNA detection method, which became the preferred clinical tool for monitoring reactivation¹⁷ (Lines 397-399 in manuscript). The transition to DNA-based assays demonstrated clear clinical advantages due to their superior sensitivity. For example, a prospective observational study comparing HCMV antigenemia and DNA monitoring found that DNA testing enabled earlier detection of reactivation¹⁸. Furthermore, it identified 67% more patients with HCMV disease who had been missed by antigenemia testing¹⁸. This earlier detection allows for earlier initiation of pre-emptive anti-HCMV therapy in HSCT patients, thereby significantly reducing the adverse outcomes¹⁹⁻²³. Despite the adoption of HCMV DNA qPCR as the standard of care, reactivation rates remain high (often > 60%)^{14, 24-28} (Lines 399-400 in manuscript), representing a persistent major challenge in HSCT patient management. Consequently, there is a clear clinical rationale for pursuing biomarkers with even greater sensitivity to enable earlier detection, with the goal of further improving

outcomes through more timely intervention.

In this study, we confirmed that the SE-SPTM-PCR platform significantly enhances the diagnostic performance of HCMV miRNAs for detecting reactivation in HSCT patients (Fig. 6 in manuscript; supplementary Fig. S25). We then conducted a retrospective cohort study involving 74 HSCT patients (NCT07181330), in whom we performed parallel HCMV miRNA (via SE-SPTM-PCR) and DNA testing on serial plasma samples. Notably, 19 patients (25.7%) were positive for HCMV miRNA prior to transplantation, despite being DNA-negative at that time. All 19 of these pre-transplant miRNA-positive patients (100%) subsequently developed DNA-positive reactivation, with a median time to DNA conversion of 15 days post-transplant (Fig. 7a-d in response letter). Among the 55 pretransplant miRNA-negative patients, 36 (65.5%) developed DNA-positive reactivation post-transplant, and 37 (67.3%) showed miRNA conversion. The median time to conversion was 22 days for DNA and 10 days for miRNA. All 36 patients who became DNA-positive also showed miRNA conversion. Importantly, in 33 of these 36 cases (92%), miRNA conversion preceded DNA conversion (Fig. 7a-d in response letter). These results collectively indicate that HCMV miRNA detection via SE-SPTM-PCR provides a significantly earlier indicator of viral activity than DNA testing. A manuscript detailing these findings is currently under review.

Encouraged by these findings, we have initiated a prospective clinical study to further evaluate the utility of HCMV miRNA for early diagnosis, therapy monitoring, and prognosis prediction in HSCT patients (NCT07210242).

Based on this trajectory of biomarker evolution and our preliminary data, we posit that HCMV miRNA detection, owing to its superior lead time and dynamic monitoring performance, has the potential to become the next-generation standard for monitoring HCMV reactivation in HSCT patients, enabling more proactive management.

Fig. 7 The results of HCMV miRNA detection and HCMV DNA detection were analyzed in the retrospective cohort of HSCT patients. **a** Analysis of test results of 74 patients. **b** Preoperative miRNA positive representative patients. **c** Representative patients with miRNA positive conversion prior to DNA positive conversion after surgery. **d** Representative patients with simultaneous positive conversion of miRNA and DNA after surgery.

4. A major practical obstacle in the clinical translation of circulating miRNA assays is the strict post-phlebotomy handling (e.g., time to centrifugation) required to prevent blood cell lysis and the subsequent artificial release of intracellular miRNAs into the plasma. This pre-analytical variability is a significant barrier to practical use. The developed SE-SPTM-PCR system is an analytical improvement but does not inherently address this pre-analytical issue. The Discussion section must acknowledge this crucial limitation. Ideally, the authors should present pilot data to demonstrate whether their selective enrichment step can mitigate the impact of delayed sample processing on their target miRNA levels.

Response: Thank you for raising this critical and insightful point regarding the pre-analytical challenges in circulating miRNA analysis. We fully agree that stringent post-phlebotomy handling requirements, particularly the risk of hemolysis and subsequent release of cellular miRNAs, represent a major barrier to the widespread clinical adoption of plasma miRNA assays. We acknowledge that the SE-SPTM-PCR platform, while an analytical advancement, does not inherently solve this fundamental pre-analytical issue. This crucial limitation has been explicitly addressed in the revised Discussion section (Lines 589-590).

Pre-analytical sample heterogeneity, such as hemolysis, can indeed affect the quantification of miRNAs in plasma. To investigate this, we prepared hemolyzed samples from freshly collected whole blood, created a serial dilution series, and analyzed them using the SE-SPTM-PCR method (Fig. 8a in response letter). Compared to non-hemolyzed plasma, the hemolyzed samples showed a significant decrease in Cq values (Fig. 8b in response letter), indicating the release of cellular miRNAs. This artificial increase demonstrates how cellular miRNA release can interfere with the accurate quantification of endogenous plasma miRNAs. Furthermore, the SE-SPTM-PCR assay could distinguish between 4-fold serial dilution steps of the hemolyzed samples, and the signal intensity was positively correlated with the dilution factor (a proxy for hemolysis severity). This suggests that, with further validation and

identification of specific miRNA indicators, the SE-SPTM-PCR platform could potentially be used to monitor pre-analytical sample quality, such as the degree of hemolysis.

To minimize pre-analytical variability, we standardized sample collection by using identical blood collection tubes, plasma separation times, and centrifugation conditions across all study groups. This protocol aimed to control for factors such as hemolysis that could affect target miRNA detection. Furthermore, to control for technical variability in nucleic acid extraction and reverse transcription, an exogenous miRNA spike-in control was added to the lysate immediately after sample lysis prior to endogenous miRNA quantification.

Fig. 8 Detection of 4-fold serially diluted hemolytic samples using SE-SPTM-PCR method. **a** 4-fold serially diluted hemolytic samples. **b** Detection results by SE-SPTM-PCR method.

5. The ROC curve shown in Figure 6d for the HCMV analysis is excessively smooth, especially given the small sample size. A proper ROC curve plotted from discrete clinical data points should be a stepped line.

Response: Thank you for this valuable suggestion. We acknowledge that the overly smooth ROC curve in Figure 6d under the condition of small sample size. Accordingly, we have increased the sample size and replotted the ROC curve using the correct stepped line format (Fig. 3c, in response letter; Figure 6d in the manuscript).

6. Figure 6b shows that conventional RT-qPCR resulted in all samples (16/16 DNA-positive and 11/11 DNA-negative) being scored as "hcmv-miR-UL22A-5p pos." If all samples were positive, how was the distinction between "positive" and "negative" made (i.e., what was the Ct or Cq cutoff value)? The authors must clearly define and justify the cutoff used to binarize the conventional RT-qPCR results for this specific miRNA.

Response: Thank you for your question. In this study, both conventional RT-qPCR and SE-SPTM-PCR assays were conducted using 45 amplification cycles. Samples exhibiting a positive amplification signal within 45 cycles were classified as positive; otherwise, they were classified as negative. We have made corresponding revisions in the Materials and Methods section regarding this aspect (Lines 648-649, 679 in the manuscript).

The results indicate that the conventional miRNA detection method exhibits poor specificity, leading to nonspecific amplification in no-template water and healthy control samples. This likely accounts for the frequent detection of HCMV miRNA in HCMV DNA-negative samples (Fig. 3a, d, g, j, m in response letter; Fig. 6b in manuscript; supplementary Fig. S25 a, d, g, j). In contrast, the SE-SPTM-PCR system demonstrated excellent specificity, with no nonspecific amplification observed in no-template water or healthy controls. This enables clear differentiation between HCMV DNA-positive and HCMV DNA-negative samples (Fig. 3b, c, e, f, h, i, k, l, n, o in response letter; Fig. 6c, d in manuscript; supplementary Fig. S25 b, c, e, f, h, i, k, l).

7. Please shorten the Introduction by focusing more concisely on the high-level challenges in circulating miRNA detection, thus improving the paper's flow and readability.

Response: We thank you for your valuable suggestions. In response, we have revised the Introduction to enhance its conciseness and to sharpen its focus on the key challenges in miRNA detection (Lines 56-146 in the manuscript).

References

1. Wu, C.W. et al. Detection of miR-92a and miR-21 in stool samples as potential screening biomarkers for colorectal cancer and polyps. *Gut* **61**, 739-745 (2012).
2. Ng, E.K. et al. Differential expression of microRNAs in plasma of patients with colorectal cancer: a potential marker for colorectal cancer screening. *Gut* **58**, 1375-1381 (2009).
3. Zheng, G. et al. Serum microRNA panel as biomarkers for early diagnosis of colorectal adenocarcinoma. *Br J Cancer* **111**, 1985-1992 (2014).
4. Hofsl, E. et al. Identification of serum microRNA profiles in colon cancer. *Br J Cancer* **108**, 1712-1719 (2013).
5. Talaya, A. et al. An investigation of the utility of plasma Cytomegalovirus (CMV) microRNA detection to predict CMV DNAemia in allogeneic hematopoietic stem cell transplant recipients. *Med Microbiol Immunol* **209**, 15-21 (2020).
6. Zhang, L., Yu, J. & Liu, Z. MicroRNAs expressed by human cytomegalovirus. *Virology* **17**, 34 (2020).
7. Zhang, G. et al. Circulating Epstein-Barr virus microRNAs miR-BART7 and miR-BART13 as biomarkers for nasopharyngeal carcinoma diagnosis and treatment. *Int J Cancer* **136**, E301-312 (2015).
8. Jiang, C. et al. Evaluation of circulating EBV microRNA BART2-5p in facilitating early detection and screening of nasopharyngeal carcinoma. *Int J Cancer* **143**, 3209-3217 (2018).
9. Rinnerthaler, G. et al. miR-16-5p Is a Stably-Expressed Housekeeping MicroRNA in Breast Cancer Tissues from Primary Tumors and from Metastatic Sites. *Int J Mol Sci* **17** (2016).
10. Pigati, L. et al. Selective release of microRNA species from normal and malignant mammary epithelial cells. *PLoS One* **5**, e13515 (2010).
11. Thu, H.N.N. et al. [miRNA-16 AS an Internal Control in Breast Cancer Studies: A Systematic Review and Meta-analysis]. *Mol Biol (Mosk)* **55**, 1045-1056 (2021).
12. Floris, I. et al. MiRNA Analysis by Quantitative PCR in Preterm Human Breast Milk

- Reveals Daily Fluctuations of hsa-miR-16-5p. *PLoS One* **10**, e0140488 (2015).
13. Manzano-Crespo, M., Atienza, M. & Cantero, J.L. Lower serum expression of miR-181c-5p is associated with increased plasma levels of amyloid-beta 1-40 and cerebral vulnerability in normal aging. *Transl Neurodegener* **8**, 34 (2019).
 14. Chen, Y. et al. Risk factors for cytomegalovirus DNAemia following haploidentical stem cell transplantation and its association with host hepatitis B virus serostatus. *J Clin Virol* **75**, 10-15 (2016).
 15. Lv, W.R. et al. Haploidentical donor transplant is associated with secondary poor graft function after allogeneic stem cell transplantation: A single-center retrospective study. *Cancer Med* **10**, 8497-8506 (2021).
 16. Chuleerarux, N. et al. The association of cytomegalovirus infection and cytomegalovirus serostatus with invasive fungal infections in allogeneic haematopoietic stem cell transplant recipients: a systematic review and meta-analysis. *Clin Microbiol Infect* **28**, 332-344 (2022).
 17. Limaye, A.P., Babu, T.M. & Boeckh, M. Progress and Challenges in the Prevention, Diagnosis, and Management of Cytomegalovirus Infection in Transplantation. *Clin Microbiol Rev* **34** (2020).
 18. Boeckh, M. et al. Optimization of quantitative detection of cytomegalovirus DNA in plasma by real-time PCR. *J Clin Microbiol* **42**, 1142-1148 (2004).
 19. Chung, H. CMV infections after HSCT: prophylaxis and treatment. *Blood Res* **60**, 33 (2025).
 20. Cui, J. et al. Diagnosis and treatment for the early stage of cytomegalovirus infection during hematopoietic stem cell transplantation. *Front Immunol* **13**, 971156 (2022).
 21. Camacho-Bydume, C. et al. Time to initiation of pre-emptive therapy for cytomegalovirus impacts overall survival in pediatric hematopoietic stem cell transplant recipients. *Cytotherapy* **24**, 428-436 (2022).
 22. Stoykova, Z., Todorova, T., Katrandzhieva, T., Kalchev, K. & Kostadinova, T. Surveillance over cytomegalovirus (CMV) reactivation following hematopoietic stem cell transplantation: a single-center experience. *Biotechnol Biotec Eq* **38** (2024).
 23. Seflek, B., Gümüs, H.H., Çimentepe, M., Küpeli, S. & Yarkin, F. Monitoring of cytomegalovirus, Epstein-Barr virus and adenovirus infections in hematopoietic stem cell transplant recipients. *Cukurova Med J* **48**, 432-440 (2023).
 24. Gao, X.N. et al. Risk factors and associations with clinical outcomes of cytomegalovirus reactivation after haploidentical versus matched-sibling unmanipulated PBSCT in patients with hematologic malignancies. *Ann Hematol* **99**, 1883-1893 (2020).
 25. Chang, Y.J. et al. Haploidentical donor is preferred over matched sibling donor for pre-transplantation MRD positive ALL: a phase 3 genetically randomized study. *J Hematol Oncol* **13**, 27 (2020).
 26. Lin, R. et al. Two dose levels of rabbit antithymocyte globulin as graft-versus-host disease prophylaxis in haploidentical stem cell transplantation: a multicenter randomized study. *BMC Med* **17**, 156 (2019).
 27. Zhang, Y.Y. et al. Comparable survival outcome between transplantation from haploidentical donor and matched related donor or unrelated donor for severe aplastic

anemia patients aged 40 years and older: A retrospective multicenter cohort study. *Clin Transplant* **34**, e13810 (2020).

28. Zhao, C. et al. Recipient and donor PTX3 rs2305619 polymorphisms increase the susceptibility to invasive fungal disease following haploidentical stem cell transplantation: a prospective study. *BMC Infect Dis* **22**, 292 (2022).

Reviewer #1 (Remarks to the Author):

The authors performed a revision according to the reviewers comments.

This is of interest for the large spectrum of readers of the journal.

Response: We sincerely appreciate your acceptance of our revision suggestions and for your positive assessment regarding the broad interest of our work. We are pleased that the revised manuscript now meets the journal's standards and appreciate you valuable time and input throughout the review process.

Reviewer #2 (Remarks to the Author):

The authors properly answered my comments.

Response: Thank you for your positive feedback on our revised manuscript. We are very pleased to learn that you found our revisions satisfactory. We appreciate the time and effort dedicated by you to evaluate our work.